# FairMerging: Rethinking Model Merging through the Lens of Fairness

**Bing Liu** [1 2 3]  **Xinrui Shan** [1]  **Boyu Zhang** [1]  **Qiankun Zhang** [1 2 3]  **Bin Yuan** [1 3 4 5 6]  **Jing Wang** [7]  **Xianjun Deng** [1 3]

## Abstract

*Model merging* offers an appealing route to multi-task learning by composing independently fine-tuned checkpoints without centralized data or retraining. However, this convenience can come with a hidden cost. Model merging may *amplify* performance disparities across subgroups, raising fairness concerns even when average accuracy remains competitive. To explain this phenomenon, we develop a sensitivity-based theoretical analysis that upper bounds the fairness gap induced by model merging. Theoretical analysis with empirical verifications reveals that the resulting fairness gap is governed by two coupled factors, a *merging magnitude* term that measures how far the merged parameters move from the target model and *global sensitivity* terms that determine how unevenly the perturbation affects subgroup losses. Guided by these insights, we propose *FairMerging*, a two-stage merging framework that first reduces the sensitivity of the target model and then performs fairness-aware coefficient optimization with orthogonally normalized task vectors. Experiments across multiple datasets, backbones, and merging baselines demonstrate that FairMerging substantially mitigates unfairness while retaining competitive multi-task performance.

## 1. Introduction

Multi-task learning (MTL) (Kollias et al., 2024; Zhang & Yang, 2021; 2018) is a machine learning paradigm that enables a model to simultaneously learn multiple related tasks through shared representations. By leveraging cross-task knowledge sharing and regularization effects, MTL improves model generalization and data efficiency, and has been widely adopted in computer vision, natural language processing, and recommendation systems. However, traditional MTL requires centralized data and joint training, which is often impractical for large foundation models due to privacy constraints, labeling and compute costs, and asynchronous task development. To address these challenges, *model merging* (Gargiulo et al., 2025; Huang et al., 2024; Yang et al., 2026; Xu et al., 2024; McMahan et al., 2017) offers a promising solution by allowing the integration of independently fine-tuned models without the need for centralized data or retraining.

Model merging composes multiple independently fine-tuned checkpoints into a single model by linearly combining their task vectors (Ilharco et al., 2023) with scalar coefficients, where each task vector is defined as the parameter difference between a fine-tuned model and a shared pretrained backbone. This data-free procedure directly reuses existing checkpoints without the need for retraining, supporting plug-and-play multi-task composition and rapid domain adaptation. The key idea behind model merging being able to achieve the purpose of MTL simply and efficiently can be attributed to two factors, the design of task vectors and the selection of merging coefficients. Most prior model merging methods focus either on shaping the task vectors or on optimizing the merging coefficients, and have achieved strong multi-task performance, as exemplified by Task Arithmetic (Ilharco et al., 2023), Ties-Merging (Yadav et al., 2023), DARE (Yu et al., 2024), AdaMerging (Yang et al., 2024), and MetaGPT (Zhou et al., 2024).

Although model merging performs well for multi-task learning, *its impact on subgroup fairness remains underexplored*. A model with strong overall accuracy can still be unreliable if it consistently underperforms for certain groups. For example, a face analysis system that works well on average but fails more often for specific age or skin-tone groups can lead to systematically unfair outcomes (Grother et al., 2019; Buolamwini & Gebru, 2018), while a traffic sign classifier that is less accurate under particular lighting or background conditions can lead to dangerous misrecognitions in driving scenarios (Lim et al., 2023; Stallkamp et al., 2012). These

---

[1]School of Cyber Science and Engineering, Huazhong University of Science and Technology, Wuhan, China [2]Key Laboratory of Cyberspace Security, Ministry of Education, Zhengzhou, China [3]Hubei Key Laboratory of Distributed System Security, Wuhan, China [4]Songshan Laboratory, Zhengzhou, China [5]Jinyinhu Laboratory, Wuhan, China [6]Visiting researcher with the Lion Rock Labs of Cyberspace Security, CTIHE, Hong Kong, China [7]School of Software Engineering, Huazhong University of Science and Technology, Wuhan, China. Correspondence to: Qiankun Zhang <qiankun@hust.edu.cn>.

*Proceedings of the 43rd International Conference on Machine Learning*, Seoul, South Korea. PMLR 306, 2026. Copyright 2026 by the author(s).

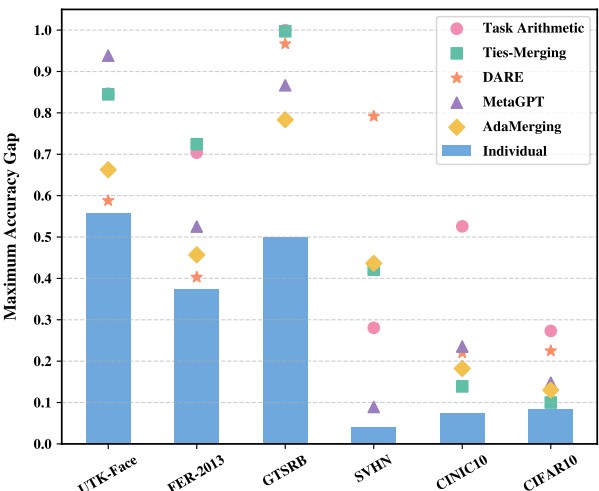

*Figure 1.* Maximum accuracy gap across subgroups before and after model merging on ViT-B/32.

concerns raise an important question: *does model merging amplify performance disparities across subgroups?* Our empirical observations suggest that the answer can be yes. Importantly, even when the individual fine-tuned models exhibit comparable subgroup gaps, merging can *amplify* these disparities in unexpected ways.

To concretely illustrate this phenomenon, we conduct a model merging study on a ViT-B/32 (Dosovitskiy et al., 2021) backbone by merging six independently fine-tuned checkpoints (UTK-Face, FER-2013, GTSRB, SVHN, CINIC10, and CIFAR10) via five representative merging baselines (Task Arithmetic, Ties-Merging, DARE, AdaMerging, and MetaGPT). We evaluate performance disparities by reporting the *maximum accuracy gap* across subgroups on each task. As shown in Figure 1, while individual checkpoints exhibit relatively moderate subgroup gaps, all five merging baselines consistently *enlarge* the maximum accuracy gaps across subgroups on every task[1]. Notably, different merging rules lead to markedly different outcomes, suggesting that fairness degradation is not merely a byproduct of multi-task interference. These observations motivate the central questions of this work: ❶ *Why does model merging lead to such subgroup disparities, and what factors govern the severity of the resulting fairness gap?* ❷ *Can we design a merging procedure that controls the fairness while retaining the multi-task performance?*

In this work, we address these questions through a sensitivity-based theoretical analysis. Our analysis reveals that the fairness impact of merging is driven by two coupled factors: a *merging magnitude* term that quantifies how far

---

[1]We replicate this phenomenon on additional backbones, including ViT-L/14 and ViT-S/16, with consistent results reported in Appendix C.1.

the merged parameters move from the target model, and *global sensitivity* terms that characterize how unevenly this parameter perturbation translates into subgroup-level performance changes (addressing ❶). In particular, the merging magnitude is governed by four determinants: the number of auxiliary tasks, the target merging coefficient, the auxiliary coefficients and task vector norms. Guided by our theoretical results, we propose FairMerging, a two-stage merging framework that mitigates subgroup unfairness while preserving multi-task performance (addressing ❷). It first reduces the global sensitivity of the target model and then performs fairness-aware coefficient optimization with orthogonally normalized task vectors to control the bound on the fairness gap derived from theoretical analysis.

In a nutshell, our contributions are summarized as follows:

- We are the first to identify that model merging can amplify subgroup performance disparities, validated through extensive empirical evidence across diverse tasks, backbones, and merging methods.

- We propose a theoretical framework that characterizes how model merging affects subgroup fairness, and identify key determinants that drive fairness amplification.

- We propose *FairMerging*, a two-stage fairness-aware model merging method. Extensive experiments on six tasks, three backbones, and five merging baselines demonstrate that FairMerging consistently reduces the subgroup fairness gap while maintaining competitive multi-task performance.

## 2. Related Work

To the best of our knowledge, there is little work reporting fairness degradation induced by model merging. We next review related work on model merging and machine learning fairness to contextualize this gap.

### 2.1. Model Merging

Model merging combines multiple independently fine-tuned checkpoints into a single multi-task model by operating directly in parameter space. Early merging approaches include simple weight averaging (Wortsman et al., 2022), as well as more principled variants such as Fisher Merging (Matena & Raffel, 2022) and RegMean (Jin et al., 2023). A major recent line of work represents each task as a task vector relative to a shared pretrained backbone and constructs a merged model by adding scaled task vectors, such as Task Arithmetic (Ilharco et al., 2023), Ties-Merging (Yadav et al., 2023), DARE (Yu et al., 2024), AdaMerging (Yang et al., 2024), and MetaGPT (Zhou et al., 2024).

Recent work has also started to examine the safety and se-

curity implications of model merging. BadMerging (Zhang et al., 2024) shows that backdoors can survive and transfer under common merging pipelines, while LoBAM (Yin et al., 2025) further highlights that such attacks remain feasible in parameter-efficient fine-tuning (e.g., LoRA) settings. Beyond demonstrating the threat, follow-up work explores defenses and mitigation, for example by constraining merging to safety-aware subspaces to reduce backdoor influence (Yang et al., 2025), and by selectively merging layers to preserve safety alignment in fine-tuned large language models (Djuhera et al., 2025). More broadly, recent evidence suggests that the risk can be compositional, as even seemingly benign components can jointly yield a backdoored merged model (Wang et al., 2025), and analogous model-hijacking behaviors have been reported for LLM merging (Yuan et al., 2025).

## 2.2. Machine Learning Fairness

Our work relates to studies that examine how machine learning techniques can reshape performance disparities across subgroups. This aligns with prior notions of fairness that emphasize auditing accuracy parity over subgroups (Hooker et al., 2020; 2019; Zafar et al., 2017), as well as evaluation practices that report worst-group or gap-based performance to surface hidden failures that average accuracy can mask (Sagawa et al., 2020; Hashimoto et al., 2018).

A growing body of work shows that post-training transformations and techniques designed for efficiency can disproportionately affect certain subgroups even when overall accuracy is largely preserved. Model compression methods such as pruning (Liu et al., 2017), quantization (Nagel et al., 2021), and low-rank adaptation (Hu et al., 2022) have been shown to induce uneven performance degradation across demographic groups (Liu et al., 2025; Ding et al., 2024; Tran et al., 2022). Beyond compression, privacy-preserving training mechanisms such as differentially private SGD (Abadi et al., 2016) can amplify subgroup accuracy gaps (Tran et al., 2021; Bagdasaryan et al., 2019). Robustness-oriented procedures exhibit similar disparities, where adversarial training and robustness auditing reveal large gaps in clean or robust accuracy across groups (Xu et al., 2021; Nanda et al., 2021).

In summary, prior work has advanced model merging for multi-task performance and explored its safety and security risks, while a separate fairness literature shows that common training and post-training procedures can unevenly affect subgroups. Our work bridges these threads by characterizing and mitigating subgroup disparities induced by task-vector-based model merging.

## 3. Preliminaries

This section formalizes the problem setup, describes the task-vector-based model merging procedure, and defines the fairness metric that serves as the basis for our analysis.

### 3.1. Problem Setting

We consider a standard supervised learning scenario in which the goal is to learn a model $f_{\boldsymbol{\theta}} : \mathcal{X} \to \mathcal{Y}$ parameterized by $\boldsymbol{\theta} \in \mathbb{R}^d$. Let $\boldsymbol{D} = \{(\boldsymbol{x}_i, y_i)\}_{i=1}^m$ denote the training dataset, where $\boldsymbol{x}_i \in \mathcal{X}$ denotes an input and $y_i \in \mathcal{Y}$ is the corresponding label. In our fairness evaluation, we treat each label as a subgroup and partition the dataset by class: $\boldsymbol{D}_a = \{(\boldsymbol{x}_i, y_i) \mid y_i = a\}, a \in \mathcal{Y}$. The training objective is to minimize the empirical risk $\mathcal{L}(\boldsymbol{\theta}; \boldsymbol{D})$ measured over the dataset $\boldsymbol{D}$:

$$\mathcal{L}(\boldsymbol{\theta}; \boldsymbol{D}) = \frac{1}{m} \sum_{i=1}^m \ell(f_{\boldsymbol{\theta}}(\boldsymbol{x}_i), y_i) , \qquad (1)$$

where $\ell : \mathcal{Y} \times \mathcal{Y} \to \mathbb{R}_+$ is a non-negative loss function such as cross-entropy.

### 3.2. Model Merging

Task-vector-based model merging is a parameter-level strategy that combines the learned knowledge of multiple models trained on different tasks or domains. Consider a set of models parameterized by weight vectors $\{\boldsymbol{\theta}_t\}_{t=0}^T$, each trained or fine-tuned independently on a different task or data distribution. These models typically share a common architecture and originate from the same pretrained model, denoted by the parameter vector $\boldsymbol{\theta}_{\mathrm{pre}}$. Formally, the task vector $\boldsymbol{\tau}_t$ for the $t$-th task is defined as the parameter difference between the task-specific model and the shared pretrained model:

$$\boldsymbol{\tau}_t = \boldsymbol{\theta}_t - \boldsymbol{\theta}_{\mathrm{pre}}, \ t = 0, 1, 2, \ldots, T . \qquad (2)$$

Then, the merged model[2] $\boldsymbol{\theta}_M$ is obtained by linearly combining the pretrained model with the scaled task vectors:

$$\boldsymbol{\theta}_M = \boldsymbol{\theta}_{\mathrm{pre}} + \sum_{t=0}^T \alpha_t \boldsymbol{\tau}_t , \qquad (3)$$

where $\alpha_t$ $(0 < \alpha_t \le 1)$ is the merging coefficient of task vector $\boldsymbol{\tau}_t$, representing the relative importance or influence on the final merged model.

### 3.3. Fairness Metric

We quantify the fairness impact of model merging by measuring how subgroup performances of the target model change after merging task vectors. Without loss of generality, we let $\boldsymbol{\theta}_0$ denote the *target model*, whose fairness

---

[2]Hereafter, unless otherwise stated, $\boldsymbol{\theta}$ denotes a model.

properties are of primary interest. The remaining models $\{\boldsymbol{\theta}_t\}_{t=1}^{T}$ are referred to as *auxiliary models*. Formally, the subgroup-level impact of merging is captured by the *excessive loss* $\mathcal{E}(a)$ for subgroup $a \in \mathcal{Y}$:

$$\mathcal{E}(a) \triangleq \mathcal{L}(\boldsymbol{\theta}_M; \boldsymbol{D}_a) - \mathcal{L}(\boldsymbol{\theta}_0; \boldsymbol{D}_a) . \quad (4)$$

To further quantify the fairness impact across all subgroups, we define the *fairness metric* $\mathcal{F}(\boldsymbol{D})$ as the maximum pairwise difference in excessive losses:

$$\mathcal{F}(\boldsymbol{D}) = \max_{a,a' \in \mathcal{Y}} |\mathcal{E}(a) - \mathcal{E}(a')| . \quad (5)$$

A higher value of $\mathcal{F}(\boldsymbol{D})$ indicates that merging has caused a larger disparity in predictive performance across subgroups, reflecting a more severe fairness gap. The following sections analyze the factors that drive increases in $\mathcal{F}(\boldsymbol{D})$ and present a fairness-aware model merging method to mitigate it.

## 4. Theoretical Fairness Analysis

In this section, we develop a theoretical framework to understand how model merging affects fairness. Building on a Taylor expansion of subgroup loss, we derive an upper bound on the excessive loss and the fairness metric, and reveal how merging rules contribute to fairness degradation.

### 4.1. Excessive Loss Bound

We begin our theoretical fairness analysis by quantifying how model merging affects the excessive loss $\mathcal{E}(a)$ on each subgroup $\boldsymbol{D}_a$. Recall that the merged model $\boldsymbol{\theta}_M$ is obtained by adding a weighted sum of task vectors to the pretrained model (Eqns. (2)–(3)). It then follows that the task vector $\boldsymbol{\tau}_M$ for the merged model is obtained by:

$$\boldsymbol{\tau}_M = \boldsymbol{\theta}_M - \boldsymbol{\theta}_{\text{pre}} = \sum_{t=0}^{T} \alpha_t \boldsymbol{\tau}_t . \quad (6)$$

To analyze the effect of merging on subgroup performance, we consider the parameter shift from the target model $\boldsymbol{\theta}_0$ to the merged model $\boldsymbol{\theta}_M$:

$$\boldsymbol{\theta}_M - \boldsymbol{\theta}_0 = \boldsymbol{\theta}_M - \boldsymbol{\theta}_{\text{pre}} + \boldsymbol{\theta}_{\text{pre}} - \boldsymbol{\theta}_0 = \boldsymbol{\tau}_M - \boldsymbol{\tau}_0 . \quad (7)$$

This key shift expresses the parameter deviation as a difference in task vectors, enabling a task-space interpretation of merging-induced perturbations. We assume that the loss function $\ell$ is twice differentiable with respect to the model parameters, which holds for common objectives such as cross-entropy and mean squared error. Under this assumption, we apply a second-order Taylor expansion of the subgroup loss $\mathcal{L}(\boldsymbol{\theta}_M; \boldsymbol{D}_a)$ around the target model $\boldsymbol{\theta}_0$, using the parameter shift derived in Eqn. (7). This yields the following result[3]:

---
[3]We defer all proofs in this paper to Appendix A.

**Theorem 4.1** (Excessive Loss Bound). *For any subgroup* $a \in \mathcal{Y}$, *the excessive loss* $\mathcal{E}(a)$ *incurred due to model merging satisfies:*

$$\begin{aligned} \mathcal{E}(a) \leq & \Delta_\alpha \cdot \left\| \boldsymbol{g}_{\boldsymbol{\theta}_0}^{D_a} \right\| + \frac{1}{2} \Delta_\alpha^2 \cdot \lambda \left( \boldsymbol{H}_{\boldsymbol{\theta}_0}^{D_a} \right) \\ & + \mathcal{O} \left( \| \boldsymbol{\tau}_M - \boldsymbol{\tau}_0 \|^3 \right) , \end{aligned} \quad (8)$$

*where* $\boldsymbol{g}_{\boldsymbol{\theta}_0}^{D_a} = \nabla_{\boldsymbol{\theta}} \mathcal{L}(\boldsymbol{\theta}; \boldsymbol{D}_a)|_{\boldsymbol{\theta}=\boldsymbol{\theta}_0}$ *denotes the gradient of the empirical loss with respect to* $\boldsymbol{\theta}_0$ *and evaluated over subgroup data* $\boldsymbol{D}_a$, $\lambda \left( \boldsymbol{H}_{\boldsymbol{\theta}_0}^{D_a} \right) = \max_i \left| \lambda_i \left( \boldsymbol{H}_{\boldsymbol{\theta}_0}^{D_a} \right) \right|$ *is the largest absolute eigenvalue of the subgroup Hessian, the term* $\mathcal{O}(\cdot)$ *denotes a negligible higher-order remainder, and the scalar* $\Delta_\alpha$ *is given by:*

$$\Delta_\alpha = (1 - \alpha_0) \cdot \| \boldsymbol{\tau}_0 \| + \sum_{t=1}^{T} \alpha_t \cdot \| \boldsymbol{\tau}_t \| . \quad (9)$$

**Implications of Theorem 4.1.** Theorem 4.1 shows that the excessive loss induced by model merging is governed by a common *merging magnitude* factor $\Delta_\alpha$ and *subgroup sensitivity* terms characterized by the gradient norm $\left\| \boldsymbol{g}_{\boldsymbol{\theta}_0}^{D_a} \right\|$ and the curvature index $\lambda \left( \boldsymbol{H}_{\boldsymbol{\theta}_0}^{D_a} \right)$. As $\Delta_\alpha$ increases, the excessive loss increases accordingly. Furthermore, the increase in $\Delta_\alpha$ *unevenly* amplifies the excessive loss across subgroups, with the degree of amplification varying depending on the differences in $\left\| \boldsymbol{g}_{\boldsymbol{\theta}_0}^{D_a} \right\|$ and $\lambda \left( \boldsymbol{H}_{\boldsymbol{\theta}_0}^{D_a} \right)$ between the subgroups. This uneven amplification enlarges the disparity in $\mathcal{E}(a)$, which in turn increases the fairness gap $\mathcal{F}(\boldsymbol{D})$. Motivated by this mechanism, we next formalize subgroup sensitivity and summarize the cross-subgroup heterogeneity via a global sensitivity measure, which allows us to derive an explicit bound on $\mathcal{F}(\boldsymbol{D})$.

### 4.2. Fairness Bound

We formalize how the fairness metric depends on subgroup sensitivity. For each subgroup $a \in \mathcal{Y}$, we quantify its *local sensitivity* to merging-induced perturbations at the target model $\boldsymbol{\theta}_0$ via the first-order (gradient) and second-order (curvature) indices:

$$G(a) = \left\| \boldsymbol{g}_{\boldsymbol{\theta}_0}^{D_a} \right\| , \quad \Lambda(a) = \lambda \left( \boldsymbol{H}_{\boldsymbol{\theta}_0}^{D_a} \right) . \quad (10)$$

By Theorem 4.1, as $\Delta_\alpha$ increases, a subgroup with larger local sensitivity $(G(a), \Lambda(a))$ is more susceptible to the influence of model merging, and experiences a greater and faster rise in excessive loss. Conversely, subgroups with smaller sensitivities are less affected and show only modest changes in excessive loss. Intuitively, larger sensitivity indices imply that the loss landscape of the subgroup around the target model is steeper and more curved, thereby rendering the

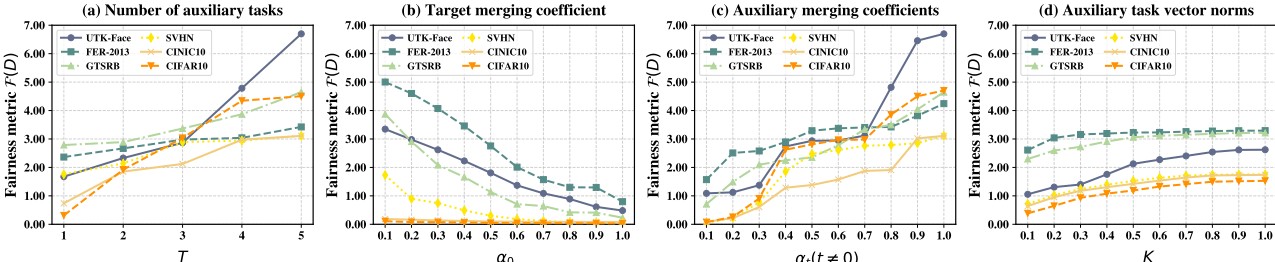

*Figure 2.* Effects of the four determinants of the merging magnitude $\Delta_\alpha$ on the fairness gap $\mathcal{F}(\boldsymbol{D})$. We use ViT-B/32 models fine-tuned on six vision benchmarks, and each curve corresponds to treating one dataset as the target task while treating the remaining datasets as auxiliary tasks. (a) Varying the number of auxiliary tasks $T$. (b–d) With all six tasks included in merging, we vary: (b) the target merging coefficient $\alpha_0$ only; (c) the auxiliary coefficients jointly, using a shared $\alpha_{t\neq 0}$ for all auxiliary task vectors; and (d) the auxiliary task vector norms via top-$K$ magnitude masking applied to each auxiliary task vector, following Ties-Merging (Yadav et al., 2023), where $K$ denotes the kept parameter ratio.

subgroup more vulnerable to the parameter perturbation induced by model merging.

To quantify how unevenly subgroups respond to the same perturbation, we define the *global sensitivity* $\left(\Delta G(\boldsymbol{D}), \Delta\Lambda(\boldsymbol{D})\right)$ as the maximum pairwise difference in local sensitivities across subgroups:

$$\Delta G(\boldsymbol{D}) = \max_{a,a'\in\mathcal{Y}} |G(a) - G(a')| \ ,$$
$$\Delta\Lambda(\boldsymbol{D}) = \max_{a,a'\in\mathcal{Y}} |\Lambda(a) - \Lambda(a')| \ . \tag{11}$$

With global sensitivity at hand, we can now translate the subgroup-wise excessive loss expansion in Theorem 4.1 into a bound on cross-subgroup disparities. Specifically, bounding the difference $\left|\mathcal{E}(a) - \mathcal{E}(a')\right|$ for any pair of subgroups $(a, a')$ in terms of $\Delta_\alpha$ and the sensitivity gaps $\left|G(a) - G(a')\right|$ and $\left|\Lambda(a) - \Lambda(a')\right|$ yields an explicit upper bound on the fairness metric $\mathcal{F}(\boldsymbol{D})$:

**Theorem 4.2** (Fairness Bound). *Let $\Delta_\alpha$ denote the merging magnitude introduced in Theorem 4.1. Then the fairness metric $\mathcal{F}(\boldsymbol{D})$ satisfies:*

$$\mathcal{F}(\boldsymbol{D}) \leq \Delta_\alpha \cdot \Delta G(\boldsymbol{D}) + \frac{1}{2}\Delta_\alpha^2 \cdot \Delta\Lambda(\boldsymbol{D}) + \mathcal{O}\left(\Delta_\alpha^3\right) \ . \tag{12}$$

**Intuition of Theorem 4.2.** Building on Theorem 4.1, Theorem 4.2 further reveals that the fairness impact of model merging is directly governed by two key factors: the *merging magnitude* $\Delta_\alpha$ and the *global sensitivity* $\left(\Delta G(\boldsymbol{D}), \Delta\Lambda(\boldsymbol{D})\right)$. The merging magnitude $\Delta_\alpha$ controls the overall amplification of merging effects, whereas $\Delta G(\boldsymbol{D})$ and $\Delta\Lambda(\boldsymbol{D})$ determine how unevenly that amplification is converted into subgroup disparities. As $\Delta_\alpha$ increases, larger $\Delta G(\boldsymbol{D})$ and $\Delta\Lambda(\boldsymbol{D})$ for the target model on dataset $\boldsymbol{D}$ are associated with more severe unfairness. The ablation studies in Section 6.3 provide further empirical evidence for this finding. This mechanism is intuitive: greater global sensitivity implies larger cross-subgroup heterogeneity in how subgroup excessive losses respond to the merging

perturbations, which in turn leads to larger performance disparities across subgroups.

### 4.3. Determinants of $\Delta_\alpha$

By Theorem 4.2, $\Delta_\alpha$ governs the overall amplification of merging-induced unfairness effects. Accordingly, understanding what determines $\Delta_\alpha$ is essential for mitigating fairness risks through merging design. From Eqn. (9), we identify four determinants. (1) **Number of auxiliary tasks** $T$. More auxiliary models increase the sum $\sum_{t=1}^{T} \alpha_t \cdot \|\boldsymbol{\tau}_t\|$, leading to a larger deviation $\Delta_\alpha$. Although additional models may offer complementary information, they also inject more deviation from the target model. (2) **Target merging coefficient** $\alpha_0$. As $\alpha_0$ decreases, the influence of the target model diminishes, which in turn increases $\Delta_\alpha$. Preserving $\alpha_0$ is thus important for fairness preservation. (3) **Auxiliary merging coefficients** $\{\alpha_t\}_{t=1}^{T}$. Larger weights assigned to auxiliary models directly increase $\Delta_\alpha$. Thus, aggressive merging policies can lead to stronger perturbations and exacerbate fairness issues. (4) **Auxiliary task vector norms** $\{\|\boldsymbol{\tau}_t\|\}_{t=1}^{T}$. Auxiliary task vectors with larger norms contribute more to $\Delta_\alpha$. Merging such models without alignment or normalization may degrade fairness.

Figure 2 illustrates how varying each of these determinants leads to an amplified fairness gap, confirming the theoretical analysis that larger $\Delta_\alpha$ values result in more severe unfairness[4]. Across all settings, we observe that increasing the number of auxiliary tasks (Figure 2(a)) and increasing either the auxiliary coefficients (Figure 2(c)) or the auxiliary task vector norms (Figure 2(d)) consistently enlarges $\mathcal{F}(\boldsymbol{D})$, whereas maintaining a larger $\alpha_0$ (Figure 2(b)) mitigates fairness degradation.

---

[4]Additional results for other ViT variants are provided in Appendix C.2.

# 5. FairMerging: Fairness-Aware Model Merging

The theoretical fairness analysis in §4 identifies the sources of unfairness in model merging, emphasizing the critical role of global sensitivity and the determinants of the merging magnitude. Building upon this understanding, we propose *FairMerging*, a fairness-aware model merging method designed to mitigate unfairness on the target task while preserving overall multi-task performance. At a high level, FairMerging proceeds in two stages. First, we perform *pre-merge fine-tuning of the target model* to directly control its global sensitivity. Second, we carry out *fairness-aware merging of task vectors*, where we (i) orthogonally normalize all task vectors to decorrelate their update directions, and (ii) optimize the merging coefficients using an entropy objective augmented with a fairness cost. Algorithm 1 and the following subsections present a detailed explanation of each step in FairMerging.

---

**Algorithm 1** Fairness-aware model merging (FairMerging)

---

**Input:** Pretrained model $\boldsymbol{\theta}_{\mathrm{pre}}$, fine-tuned models $\{\boldsymbol{\theta}_t\}_{t=0}^T$.
**Output:** Merged model $\boldsymbol{\theta}_M$.
1: **// Stage 1: Fine-tune the target model $\theta_0$ by solving the optimization in Eqn. (13).**
2: $\tilde{\boldsymbol{\theta}}_0 \Leftarrow \arg\min_{\boldsymbol{\theta}} \tilde{\mathcal{L}}(\boldsymbol{\theta}; \boldsymbol{D})$;
3: **//** Create task vectors w.r.t. $\boldsymbol{\theta}_{\mathrm{pre}}$
4: $\tilde{\boldsymbol{\tau}}_0 \Leftarrow \tilde{\boldsymbol{\theta}}_0 - \boldsymbol{\theta}_{\mathrm{pre}}$;
5: $\boldsymbol{\tau}_t \Leftarrow \boldsymbol{\theta}_t - \boldsymbol{\theta}_{\mathrm{pre}} \ (t = 1, 2, \dots, T)$;
6: **// Stage 2.1: Orthogonally normalize the task vectors.**
7: $\tilde{\boldsymbol{\tau}}_t \Leftarrow \boldsymbol{\tau}_t - \sum_{i=0}^{t-1} \frac{\langle \boldsymbol{\tau}_t, \tilde{\boldsymbol{\tau}}_i \rangle}{\langle \tilde{\boldsymbol{\tau}}_i, \tilde{\boldsymbol{\tau}}_i \rangle} \tilde{\boldsymbol{\tau}}_i \ (t = 1, 2, \dots, T)$;
8: $\hat{\boldsymbol{\tau}}_t \Leftarrow \frac{\tilde{\boldsymbol{\tau}}_t}{\|\tilde{\boldsymbol{\tau}}_t\|} \ (t = 1, 2, \dots, T)$;
9: $\hat{\boldsymbol{\tau}}_0 \Leftarrow \tilde{\boldsymbol{\tau}}_0$;
10: **// Stage 2.2: Optimize the merging coefficients in a fairness-aware manner.**
11: **//** Compute the global sensitivity of target model $\tilde{\boldsymbol{\theta}}_0$
12: $\tilde{\Delta}G(\boldsymbol{D}) \Leftarrow \max_{a,a' \in \mathcal{Y}} \left| \left\| \boldsymbol{g}_{\tilde{\boldsymbol{\theta}}_0}^{\boldsymbol{D}_a} \right\| - \left\| \boldsymbol{g}_{\tilde{\boldsymbol{\theta}}_0}^{\boldsymbol{D}_{a'}} \right\| \right|$;
13: $\tilde{\Delta}\Lambda(\boldsymbol{D}) \Leftarrow \max_{a,a' \in \mathcal{Y}} \left| \lambda\left( \boldsymbol{H}_{\tilde{\boldsymbol{\theta}}_0}^{\boldsymbol{D}_a} \right) - \lambda\left( \boldsymbol{H}_{\tilde{\boldsymbol{\theta}}_0}^{\boldsymbol{D}_{a'}} \right) \right|$;
14: **//** Optimize the merging coefficients by solving the optimization in Eqn. (20), according to Eqns. (17) and (19).
15: $\hat{\boldsymbol{\alpha}} \Leftarrow \arg\min_{\boldsymbol{\alpha}} \left\{ \mathcal{L}_{\mathrm{ent}}(\boldsymbol{\alpha}) + \eta \cdot J_{\mathrm{fair}}(\boldsymbol{\alpha}) \right\}$;
16: **//** Obtain the merged model.
17: $\boldsymbol{\theta}_M \Leftarrow \boldsymbol{\theta}_{\mathrm{pre}} + \sum_{t=0}^{T} \hat{\alpha}_t \hat{\boldsymbol{\tau}}_t$.

---

## 5.1. Pre-Merge Fine-Tuning of the Target Model

The first stage in the FairMerging process is to fine-tune the target model before merging to control its global sensitivity.

The fine-tuned target model $\tilde{\boldsymbol{\theta}}_0$ is obtained by solving:

$$
\min_{\boldsymbol{\theta}} \tilde{\mathcal{L}}(\boldsymbol{\theta}; \boldsymbol{D}) = \underbrace{\sum_{a \in \mathcal{Y}} \mathcal{L}(\boldsymbol{\theta}; \boldsymbol{D}_a)}_{\text{subgroup loss}}
$$

$$
+ \underbrace{\mu_G \cdot \mathrm{Var}_{a \in \mathcal{Y}}\left[ G(a) \right] + \mu_\Lambda \cdot \mathrm{Var}_{a \in \mathcal{Y}}\left[ \Lambda(a) \right]}_{\text{global sensitivity control}}
$$

$$
+ \underbrace{\rho \cdot \|\boldsymbol{\theta} - \boldsymbol{\theta}_0\|^2}_{\text{regularization}} .
$$

$$\tag{13}$$

In this optimization, the objective $\tilde{\mathcal{L}}(\boldsymbol{\theta}; \boldsymbol{D})$ comprises three components:

- **Subgroup loss.** By the definition in Eqn. (1), $\sum_{a \in \mathcal{Y}} \mathcal{L}(\boldsymbol{\theta}; \boldsymbol{D}_a)$ represents the weighted sum of subgroup losses with weights $\frac{1}{|\boldsymbol{D}_a|}$, which eliminates the bias caused by sample-size imbalance.

- **Global sensitivity control.** $\mathrm{Var}_{a \in \mathcal{Y}}\left[ G(a) \right]$ and $\mathrm{Var}_{a \in \mathcal{Y}}\left[ \Lambda(a) \right]$ denote, respectively, the across-subgroup variance of the first-order and second-order sensitivity indices. Compared with the max-gap measure in Eqn. (11), a variance-based global sensitivity surrogate is smooth and differentiable, which enables more stable, computationally efficient optimization. By narrowing these variances, the method lowers the global sensitivity terms that drive fairness gaps in our theory. Hyperparameters $\mu_G \geq 0$ and $\mu_\Lambda \geq 0$ modulate the accuracy-fairness trade-off.

- **Regularization.** The term $\rho \cdot \|\boldsymbol{\theta} - \boldsymbol{\theta}_0\|^2$ constrains deviations from the original target model to avoid overfitting and preserve the target model's baseline accuracy.

## 5.2. Fairness-Aware Merging of Task Vectors

Guided by the analysis of Theorem 4.2, the second stage first orthogonally normalizes the task vectors and then performs fairness-aware optimization of the merging coefficients.

**Orthogonal normalization of task vectors.** To reduce redundancy among tasks and prevent correlated updates from compounding fairness risk, we decorrelate the task vectors. Concretely, we apply a Gram–Schmidt transform over the set $\{\tilde{\boldsymbol{\tau}}_0, \boldsymbol{\tau}_1, \boldsymbol{\tau}_2, \dots, \boldsymbol{\tau}_T\}$, projecting each task vector onto the subspace orthogonal to those already processed:

$$
\tilde{\boldsymbol{\tau}}_t = \boldsymbol{\tau}_t - \sum_{i=0}^{t-1} \frac{\langle \boldsymbol{\tau}_t, \tilde{\boldsymbol{\tau}}_i \rangle}{\langle \tilde{\boldsymbol{\tau}}_i, \tilde{\boldsymbol{\tau}}_i \rangle} \tilde{\boldsymbol{\tau}}_i \quad (t = 1, 2, \dots, T), \tag{14}
$$

where the target task vector $\tilde{\boldsymbol{\tau}}_0 = \tilde{\boldsymbol{\theta}}_0 - \boldsymbol{\theta}_{\text{pre}}$. We then rescale each resulting vector to unit length:

$$\hat{\boldsymbol{\tau}}_t = \frac{\tilde{\boldsymbol{\tau}}_t}{\|\tilde{\boldsymbol{\tau}}_t\|} \quad (t = 1, 2, \ldots, T) . \tag{15}$$

**Fairness-aware optimization of merging coefficients.**
Given the orthogonally normalized task vectors $\{\hat{\boldsymbol{\tau}}_t\}_{t=1}^T$ and the target task vector $\hat{\boldsymbol{\tau}}_0 = \tilde{\boldsymbol{\tau}}_0$, the merged model is parameterized by the coefficients $\boldsymbol{\alpha} = (\alpha_0, \ldots, \alpha_T)$ as

$$\boldsymbol{\theta}_M(\boldsymbol{\alpha}) = \boldsymbol{\theta}_{\text{pre}} + \sum_{t=0}^T \alpha_t \hat{\boldsymbol{\tau}}_t . \tag{16}$$

The final step of FairMerging optimizes these coefficients in a way that balances overall performance and the fairness bound in Theorem 4.2. To this end, we combine the unsupervised entropy objective of AdaMerging (Yang et al., 2024) with a fairness cost on the theoretical bound.

Following AdaMerging, we treat $\boldsymbol{\alpha}$ as continuous parameters and use unlabeled data from all tasks to define an uncertainty-based objective:

$$\mathcal{L}_{\text{ent}}(\boldsymbol{\alpha}) = \sum_{t=0}^T \sum_{\boldsymbol{x} \in \mathcal{U}_t} H\big(f_{\boldsymbol{\theta}_M(\boldsymbol{\alpha})}(\boldsymbol{x})\big) , \tag{17}$$

where $\mathcal{U}_t$ denotes an unlabeled dataset from task $t$ and $H(\cdot)$ is the Shannon entropy (Shannon, 1948) of the prediction. Minimizing $\mathcal{L}_{\text{ent}}$ encourages the merged model to produce confident predictions on all tasks, and empirically serves as an effective proxy for improving multi-task accuracy.

On top of this performance-oriented term, we incorporate a fairness-aware penalty motivated by the theoretical bound in Theorem 4.2. After orthogonal normalization, all auxiliary task vectors have a unit norm, so the effective merging magnitude is fully controlled by

$$\tilde{\Delta}_\alpha = (1 - \alpha_0) \cdot \|\hat{\boldsymbol{\tau}}_0\| + \sum_{t=1}^T \alpha_t . \tag{18}$$

Let $\tilde{\Delta}G(\boldsymbol{D})$ and $\tilde{\Delta}\Lambda(\boldsymbol{D})$ be the global sensitivity indices computed on the pre-merge fine-tuned target model $\tilde{\boldsymbol{\theta}}_0$ over the target dataset $\boldsymbol{D}$. Inspired by the fairness bound in Theorem 4.2, we define the fairness cost

$$J_{\text{fair}}(\boldsymbol{\alpha}) = \tilde{\Delta}_\alpha \cdot \tilde{\Delta}G(\boldsymbol{D}) + \frac{1}{2}\tilde{\Delta}_\alpha^2 \cdot \tilde{\Delta}\Lambda(\boldsymbol{D}) . \tag{19}$$

Combining the entropy objective with the fairness cost yields our final optimization problem:

$$\hat{\boldsymbol{\alpha}} = \arg\min_{\boldsymbol{\alpha}} \Big\{ \mathcal{L}_{\text{ent}}(\boldsymbol{\alpha}) + \eta \cdot J_{\text{fair}}(\boldsymbol{\alpha}) \Big\}, \tag{20}$$

where $\eta \geq 0$ controls the trade-off between overall multi-task performance and fairness on the target task.

**Overhead costs.** FairMerging introduces two main sources of overhead: (i) Stage 1 performs an additional fine-tuning step on the target task, which requires labeled target-task data and subgroup annotations; and (ii) Stage 2 optimizes the merging coefficients using unlabeled data from all tasks, which introduces additional optimization cost. However, this overhead is moderate in practice: the main additional cost is confined to Stage 1, which only requires 1–5 epochs of fine-tuning, whereas Stage 2 is comparable in spirit and scale to AdaMerging, which also relies on unlabeled entropy minimization.

## 6. Experiments

In this section, we evaluate whether FairMerging can mitigate unfairness while maintaining competitive multi-task performance, and we further conduct ablation studies to quantify the contribution of each FairMerging component.

### 6.1. Experimental Setup

**Datasets and models.** Following prior task-vector-based merging setups (Yang et al., 2024; Ilharco et al., 2023), we study multi-task model merging on six image classification datasets: UTK-Face (Zhang et al., 2017), FER-2013 (Goodfellow et al., 2013), GTSRB (Stallkamp et al., 2011), SVHN (Netzer et al., 2011), CINIC10 (Darlow et al., 2018), and CIFAR10 (Krizhevsky et al., 2009). We provide a more detailed description of the datasets in Appendix B.1. Unless otherwise stated, all task-specific models are independently fine-tuned from the same pretrained ViT-B/32 backbone. We additionally verify the main conclusions on ViT-L/14 and ViT-S/16 (Touvron et al., 2021), with results reported in Appendix C.3.

**Baselines and metrics.** We compare FairMerging against five representative task-vector-based model merging baselines, including Task Arithmetic (Ilharco et al., 2023), Ties-Merging (Yadav et al., 2023), DARE (Yu et al., 2024), AdaMerging (Yang et al., 2024), and MetaGPT (Zhou et al., 2024). For subgroup fairness on the target task, we report the fairness metric $\mathcal{F}(\boldsymbol{D})$. For multi-task performance, we report the test classification accuracy on each task and the average accuracy (i.e., Avg Acc) across six tasks. Additional implementation details and baseline configurations are provided in Appendix B.2.

### 6.2. Fairness and MTL Performance

We evaluate FairMerging from two key perspectives, focusing on subgroup fairness on the target task and overall multi-task performance across all tasks.

**Substantially mitigated unfairness.** Table 1 reports the fairness metric $\mathcal{F}(\boldsymbol{D})$ (lower is better) on the target task

*Table 1.* Fairness metric $\mathcal{F}(\boldsymbol{D})$ when merging ViT-B/32 models on six tasks.

| Method | UTK-Face | FER-2013 | GTSRB | SVHN | CINIC10 | CIFAR10 | Avg |
|---|---|---|---|---|---|---|---|
| Task Arithmetic (Ilharco et al., 2023) | 3.28 | 3.05 | 4.02 | 0.71 | 1.10 | 0.70 | 2.14 |
| Ties-Merging (Yadav et al., 2023) | 2.52 | 4.02 | 3.97 | 1.69 | 0.24 | 0.21 | 2.11 |
| DARE (Yu et al., 2024) | 3.71 | 2.89 | 3.02 | 3.83 | 0.37 | 0.47 | 2.38 |
| MetaGPT (Zhou et al., 2024) | 3.15 | 2.87 | 2.11 | 0.81 | 0.68 | 0.32 | 1.66 |
| AdaMerging (Yang et al., 2024) | 2.62 | 2.49 | 2.04 | 1.31 | 0.31 | 0.18 | 1.49 |
| FairMerging (Full) | **0.57** | **0.70** | **0.62** | **0.48** | **0.07** | **0.09** | **0.42** |
| *Ablation Studies* | | | | | | | |
| FairMerging (Stage 1) + Task Arithmetic | 3.03 | 2.69 | 2.87 | 0.63 | 0.89 | 0.56 | 1.78 |
| FairMerging (Stage 1) + Ties-Merging | 2.23 | 3.95 | 3.06 | 1.53 | 0.19 | 0.12 | 1.85 |
| FairMerging (Stage 1) + DARE | 3.36 | 2.63 | 2.75 | 3.69 | 0.25 | 0.34 | 2.17 |
| FairMerging (Stage 1) + MetaGPT | 2.97 | 2.48 | 1.81 | 0.70 | 0.47 | 0.15 | 1.43 |
| FairMerging (Stage 1) + AdaMerging | 2.25 | 2.13 | 1.68 | 1.13 | 0.19 | 0.14 | 1.26 |
| FairMerging (Stage 2) | **0.84** | **0.98** | **0.81** | **0.63** | **0.12** | **0.10** | **0.58** |

*Table 2.* Multi-task performance (%) when merging ViT-B/32 models on six tasks.

| Method | UTK-Face | FER-2013 | GTSRB | SVHN | CINIC10 | CIFAR10 | Avg Acc |
|---|---|---|---|---|---|---|---|
| Pretrained | 42.20 | 16.69 | 46.97 | 59.73 | 53.89 | 67.30 | 47.80 |
| Individual | 83.96 | 67.74 | 98.67 | 97.05 | 95.06 | 97.03 | 89.92 |
| Task Arithmetic (Ilharco et al., 2023) | 48.68 | 18.92 | 69.81 | 85.84 | 72.18 | 86.32 | 63.63 |
| Ties-Merging (Yadav et al., 2023) | 74.64 | 50.77 | 69.59 | 50.27 | 90.82 | **97.38** | 72.24 |
| DARE (Yu et al., 2024) | 67.97 | 41.49 | 86.18 | 56.86 | 85.75 | 93.26 | 71.92 |
| MetaGPT (Zhou et al., 2024) | 57.22 | 25.10 | 89.31 | **95.16** | 83.09 | 93.35 | 73.87 |
| AdaMerging (Yang et al., 2024) | **75.84** | 51.14 | **90.48** | 71.09 | 90.44 | 96.53 | **79.25** |
| FairMerging (Full) | 69.87 | **52.75** | 73.87 | 65.96 | **92.15** | 95.62 | 75.04 |

under different merging rules. Across all six tasks, existing merging baselines incur noticeably larger fairness gaps, indicating that task-vector-based model merging can systematically amplify subgroup disparities. In contrast, FairMerging consistently achieves the lowest average fairness gap, reducing the average $\mathcal{F}(\boldsymbol{D})$ from the best baseline value of 1.49 (AdaMerging) to 0.42, with improvements observed on every dataset (e.g., UTK-Face: $2.62 \rightarrow 0.57$; FER-2013: $2.49 \rightarrow 0.70$; GTSRB: $2.04 \rightarrow 0.62$). These results support our thesis that merging-induced fairness degradation can be controlled by regulating the effective merging magnitude and the global sensitivity characterized in Theorem 4.2.

**Competitive multi-task performance.** Table 2 reports the per-task classification accuracy and the average accuracy across six tasks. FairMerging achieves an average accuracy of 75.04%, approaching close to the strongest performance-oriented baseline, AdaMerging (Avg Acc 79.25%), while outperforming all other merging baselines. At the per-task level, FairMerging achieves the best performance on FER-2013 (52.75%) and CINIC10 (92.15%), indicating that FairMerging can retain strong task-level accuracy while en-

forcing fairness-aware merging.

### 6.3. Ablation Studies

We ablate FairMerging to quantify how each stage contributes to mitigating fairness degradation, with results summarized in Table 1.

**The role of global sensitivity control.** We examine Stage 1 by applying it as a preprocessing step to the target model and then performing coefficient selection using standard merging baselines. Concretely, we combine the Stage 1 fine-tuned target model with each of the five baselines, yielding variants such as **Stage 1 + Task Arithmetic**. Across tasks, these variants consistently reduce the fairness metric relative to their original counterparts, for instance lowering the average $\mathcal{F}(\boldsymbol{D})$ from 2.14 to 1.78 for Task Arithmetic and from 1.49 to 1.26 for AdaMerging. This pattern indicates that part of the unfairness amplification is already determined by the target model's sensitivity, and that controlling global sensitivity offers a preliminary way to mitigate fairness degradation. Furthermore, these findings provide empirical

support for Theorem 4.2, which predicts that larger global sensitivity terms lead to a larger fairness bound.

**The role of fairness-aware merging.** We next evaluate the contribution of Stage 2 by applying it directly to the target model without Stage 1. Compared with performance-driven merging baselines, Stage 2 significantly reduces the fairness metric, achieving an average $\mathcal{F}(\boldsymbol{D})$ of 0.58. This improvement is consistent with Theorem 4.2, since Stage 2 explicitly penalizes the bound-inspired fairness cost. Moreover, Stage 2 yields a larger improvement than Stage 1, indicating that fairness-aware optimization of merging coefficients is more effective than sensitivity control alone.

Due to the page limitations, we further provide additional ablations on the key modules of FairMerging and the fairness regularization coefficients $\eta$, $\mu_G$, and $\mu_\Lambda$ in Appendix C.4-C.7. We also include preliminary NLP experiments in Appendix C.8 to further examine the generalizability of our findings.

## 7. Conclusion

Motivated by consistent empirical observations across multiple datasets and backbones, we find that task-vector-based model merging can systematically amplify subgroup performance disparities. To the best of our knowledge, this work is the first to formally characterize this phenomenon both theoretically and empirically, and to propose a fairness-aware model merging framework to address it. A limitation of FairMerging is that its sensitivity-control stage relies on labeled target-task data and subgroup annotations. Further studying its behavior under severe label scarcity, noisy subgroup annotations, or unavailable subgroup information is an important direction for future work.

## Acknowledgements

Qiankun Zhang is supported by the National Natural Science Foundation of China (Grant 62302183), Open Foundation of Key Laboratory of Cyberspace Security, Ministry of Education of China (Grant KLCS20240401), and CCF-DiDi GAIA Collaborative Research Funds (Grant CCF-DiDi GAIA 202522). Jing Wang is supported by the National Natural Science Foundation of China (Grant 62572208), and the Nature Science Foundation of Hubei Province (Grant 2025DJA050, 2025AFD748). Bin Yuan is supported by the National Natural Science Foundation of China (Grant 62372191), the Open Topics from The Lion Rock Labs of Cyberspace Security (Grant LRL24013), and Songshan Laboratory (Grant 241110210200). Xianjun Deng is supported by the National Key R&D Program of China (Grant 2022YFE0138600), and the National Natural Science Foundation of China (Grant U24B20153).

## Impact Statement

This paper presents work whose goal is to advance the field of Machine Learning. There are many potential societal consequences of our work, none of which we feel must be specifically highlighted here.

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

# A. Missing Proofs

## A.1. Proof of Theorem 4.1

*Proof.* **Step 1 (Merge-induced task vector shift).** By Eqn. (6) we have $\boldsymbol{\tau}_M = \sum_{t=0}^{T} \alpha_t \boldsymbol{\tau}_t$. Subtracting the target task vector $\boldsymbol{\tau}_0$ from both sides and isolating the $t = 0$ term yields:

$$\boldsymbol{\tau}_M - \boldsymbol{\tau}_0 = \sum_{t=0}^{T} \alpha_t \boldsymbol{\tau}_t - \boldsymbol{\tau}_0 = (\alpha_0 - 1)\boldsymbol{\tau}_0 + \sum_{t=1}^{T} \alpha_t \boldsymbol{\tau}_t , \tag{21}$$

which is the desired identity.

**Step 2 (Second–order Taylor expansion around the target).** By the assumption that the loss function $\ell$ is twice differentiable, the subgroup empirical loss $\mathcal{L}(\boldsymbol{\theta}; \boldsymbol{D}_a)$ is twice continuously differentiable with respect to $\boldsymbol{\theta}$ on a neighborhood of $\boldsymbol{\theta}_0$. Consequently, a second–order Taylor expansion of $\mathcal{L}(\boldsymbol{\theta}_M; \boldsymbol{D}_a)$ at $\boldsymbol{\theta}_0$ gives:

$$\begin{aligned}
\mathcal{L}(\boldsymbol{\theta}_M; \boldsymbol{D}_a) &= \mathcal{L}\left((\boldsymbol{\theta}_M - \boldsymbol{\theta}_0) + \boldsymbol{\theta}_0; \boldsymbol{D}_a\right) \\
&= \mathcal{L}(\boldsymbol{\theta}_0; \boldsymbol{D}_a) + (\boldsymbol{\theta}_M - \boldsymbol{\theta}_0)^{\mathrm{T}} \boldsymbol{g}_{\boldsymbol{\theta}_0}^{D_a} + \frac{1}{2}(\boldsymbol{\theta}_M - \boldsymbol{\theta}_0)^{\mathrm{T}} \boldsymbol{H}_{\boldsymbol{\theta}_0}^{D_a}(\boldsymbol{\theta}_M - \boldsymbol{\theta}_0) + \mathcal{O}\left(\|\boldsymbol{\theta}_M - \boldsymbol{\theta}_0\|^3\right) ,
\end{aligned} \tag{22}$$

where $\boldsymbol{g}_{\boldsymbol{\theta}_0}^{D_a} = \nabla_{\boldsymbol{\theta}} \mathcal{L}(\boldsymbol{\theta}; \boldsymbol{D}_a)\big|_{\boldsymbol{\theta}=\boldsymbol{\theta}_0}$ is the gradient and $\boldsymbol{H}_{\boldsymbol{\theta}_0}^{D_a} = \nabla_{\boldsymbol{\theta}}^2 \mathcal{L}(\boldsymbol{\theta}; \boldsymbol{D}_a)\big|_{\boldsymbol{\theta}=\boldsymbol{\theta}_0}$ is the Hessian at $\boldsymbol{\theta}_0$. The term $\mathcal{O}\left(\|\boldsymbol{\theta}_M - \boldsymbol{\theta}_0\|^3\right)$ denotes a negligible higher-order remainder.

**Step 3 (Deriving the excessive loss bound).** The excessive loss $\mathcal{E}(a)$ for subgroup $a \in \mathcal{Y}$ is given by definition in Eqn. (4):

$$\mathcal{E}(a) = \mathcal{L}(\boldsymbol{\theta}_M; \boldsymbol{D}_a) - \mathcal{L}(\boldsymbol{\theta}_0; \boldsymbol{D}_a) \tag{23a}$$

$$= (\boldsymbol{\theta}_M - \boldsymbol{\theta}_0)^{\mathrm{T}} \boldsymbol{g}_{\boldsymbol{\theta}_0}^{D_a} + \frac{1}{2}(\boldsymbol{\theta}_M - \boldsymbol{\theta}_0)^{\mathrm{T}} \boldsymbol{H}_{\boldsymbol{\theta}_0}^{D_a}(\boldsymbol{\theta}_M - \boldsymbol{\theta}_0) + \mathcal{O}\left(\|\boldsymbol{\theta}_M - \boldsymbol{\theta}_0\|^3\right) \tag{23b}$$

$$= (\boldsymbol{\tau}_M - \boldsymbol{\tau}_0)^{\mathrm{T}} \boldsymbol{g}_{\boldsymbol{\theta}_0}^{D_a} + \frac{1}{2}(\boldsymbol{\tau}_M - \boldsymbol{\tau}_0)^{\mathrm{T}} \boldsymbol{H}_{\boldsymbol{\theta}_0}^{D_a}(\boldsymbol{\tau}_M - \boldsymbol{\tau}_0) + \mathcal{O}\left(\|\boldsymbol{\tau}_M - \boldsymbol{\tau}_0\|^3\right) \tag{23c}$$

$$\leq \|\boldsymbol{\tau}_M - \boldsymbol{\tau}_0\| \cdot \left\|\boldsymbol{g}_{\boldsymbol{\theta}_0}^{D_a}\right\| + \frac{1}{2} \cdot \|\boldsymbol{\tau}_M - \boldsymbol{\tau}_0\|^2 \cdot \left\|\boldsymbol{H}_{\boldsymbol{\theta}_0}^{D_a}\right\| + \mathcal{O}\left(\|\boldsymbol{\tau}_M - \boldsymbol{\tau}_0\|^3\right) \tag{23d}$$

$$= \left\|(\alpha_0 - 1)\boldsymbol{\tau}_0 + \sum_{t=1}^{T} \alpha_t \boldsymbol{\tau}_t\right\| \cdot \left\|\boldsymbol{g}_{\boldsymbol{\theta}_0}^{D_a}\right\| + \frac{1}{2} \cdot \left\|(\alpha_0 - 1)\boldsymbol{\tau}_0 + \sum_{t=1}^{T} \alpha_t \boldsymbol{\tau}_t\right\|^2 \cdot \left\|\boldsymbol{H}_{\boldsymbol{\theta}_0}^{D_a}\right\| + \mathcal{O}\left(\|\boldsymbol{\tau}_M - \boldsymbol{\tau}_0\|^3\right) \tag{23e}$$

$$\leq \left((1 - \alpha_0) \cdot \|\boldsymbol{\tau}_0\| + \sum_{t=1}^{T} \alpha_t \cdot \|\boldsymbol{\tau}_t\|\right) \cdot \left\|\boldsymbol{g}_{\boldsymbol{\theta}_0}^{D_a}\right\| + \frac{1}{2} \cdot \left((1 - \alpha_0) \cdot \|\boldsymbol{\tau}_0\| + \sum_{t=1}^{T} \alpha_t \cdot \|\boldsymbol{\tau}_t\|\right)^2 \cdot \left\|\boldsymbol{H}_{\boldsymbol{\theta}_0}^{D_a}\right\|$$
$$+ \mathcal{O}\left(\|\boldsymbol{\tau}_M - \boldsymbol{\tau}_0\|^3\right) \tag{23f}$$

$$= \left((1 - \alpha_0) \cdot \|\boldsymbol{\tau}_0\| + \sum_{t=1}^{T} \alpha_t \cdot \|\boldsymbol{\tau}_t\|\right) \cdot \left\|\boldsymbol{g}_{\boldsymbol{\theta}_0}^{D_a}\right\|$$
$$+ \frac{1}{2} \cdot \left((1 - \alpha_0) \cdot \|\boldsymbol{\tau}_0\| + \sum_{t=1}^{T} \alpha_t \cdot \|\boldsymbol{\tau}_t\|\right)^2 \cdot \max_i \left|\lambda_i\left(\boldsymbol{H}_{\boldsymbol{\theta}_0}^{D_a}\right)\right| + \mathcal{O}\left(\|\boldsymbol{\tau}_M - \boldsymbol{\tau}_0\|^3\right) \tag{23g}$$

$$= \Delta_\alpha \cdot \left\|\boldsymbol{g}_{\boldsymbol{\theta}_0}^{D_a}\right\| + \frac{1}{2} \Delta_\alpha^2 \cdot \lambda\left(\boldsymbol{H}_{\boldsymbol{\theta}_0}^{D_a}\right) + \mathcal{O}\left(\|\boldsymbol{\tau}_M - \boldsymbol{\tau}_0\|^3\right) , \tag{23h}$$

where $\lambda\left(\boldsymbol{H}_{\boldsymbol{\theta}_0}^{D_a}\right) = \max_i \left|\lambda_i\left(\boldsymbol{H}_{\boldsymbol{\theta}_0}^{D_a}\right)\right|$ and $\Delta_\alpha = (1 - \alpha_0) \cdot \|\boldsymbol{\tau}_0\| + \sum_{t=1}^{T} \alpha_t \cdot \|\boldsymbol{\tau}_t\|$.

Eqn. (23b) follows from the second–order Taylor expansion in Eqn. (22). Eqn. (23c) is obtained by Eqn. (7). Eqn. (23d) follows by applying the Cauchy-Schwarz inequality to the linear term and the Rayleigh-Ritz (operator-norm) bound to the quadratic term. Eqn. (23e) results from the merge-induced task vector shift in Eqn. (21). Eqn. (23f) follows from the triangle inequality for the Euclidean norm. Eqn. (23g) from the fact that, for a real symmetric matrix $H$, the operator norm equals the largest absolute eigenvalue, i.e., $\|H\| = \max_i |\lambda_i(H)|$. $\square$

## A.2. Proof of Theorem 4.2

*Proof.* We begin by introducing the merging–weighted proxy for the subgroup excessive loss:

$$S_\Delta(a) \triangleq \Delta_\alpha \cdot G(a) + \frac{1}{2}\Delta_\alpha^2 \cdot \Lambda(a) \,. \tag{24}$$

By Theorem 4.1 and Taylor's theorem with integral remainder, there exist constants $C_a, C_{a'} > 0$ such that, for any fixed $a, a' \in \mathcal{Y}$:

$$\mathcal{E}(a) \le S_\Delta(a) + C_a\,\Delta_\alpha^3, \qquad \mathcal{E}(a') \le S_\Delta(a') + C_{a'}\,\Delta_\alpha^3 \,. \tag{25}$$

Subtracting the second inequality from the first yields:

$$\mathcal{E}(a) - \mathcal{E}(a') \le S_\Delta(a) - S_\Delta(a') + (C_a + C_{a'})\Delta_\alpha^3 \,. \tag{26}$$

Swapping $(a, a')$ and repeating gives:

$$\mathcal{E}(a') - \mathcal{E}(a) \le S_\Delta(a') - S_\Delta(a) + (C_a + C_{a'})\Delta_\alpha^3 \,, \tag{27}$$

which is equivalent to:

$$\mathcal{E}(a) - \mathcal{E}(a') \ge S_\Delta(a) - S_\Delta(a') - (C_a + C_{a'})\Delta_\alpha^3 \,. \tag{28}$$

Combining Eqn. (26) and Eqn. (28) together yields:

$$|\mathcal{E}(a) - \mathcal{E}(a')| \le |S_\Delta(a) - S_\Delta(a')| + (C_a + C_{a'})\Delta_\alpha^3 = |S_\Delta(a) - S_\Delta(a')| + \mathcal{O}(\Delta_\alpha^3) \,. \tag{29}$$

Next, by the definition in Eqn. (24):

$$|S_\Delta(a) - S_\Delta(a')| \le \Delta_\alpha \cdot |G(a) - G(a')| + \frac{1}{2}\Delta_\alpha^2 \cdot |\Lambda(a) - \Lambda(a')| \,. \tag{30}$$

Taking the maximum over $(a, a')$ and applying the global sensitivity in Eqn. (11), we obtain:

$$\mathcal{F}(\boldsymbol{D}) = \max_{a,a' \in \mathcal{Y}} |\mathcal{E}(a) - \mathcal{E}(a')| \le \Delta_\alpha \cdot \Delta G(\boldsymbol{D}) + \frac{1}{2}\Delta_\alpha^2 \cdot \Delta\Lambda(\boldsymbol{D}) + \mathcal{O}\left(\Delta_\alpha^3\right) \,. \tag{31}$$

$\square$

# B. Experimental Settings

This section provides detailed descriptions of the datasets and baselines, along with training details.

## B.1. Datasets Details

We evaluate FairMerging on six widely used image classification benchmarks that cover facial analysis, emotion recognition, traffic sign recognition, digit recognition, and object recognition.

- **UTK-Face** (Zhang et al., 2017) is a face dataset annotated with age, gender, and ethnicity, and is commonly used for fairness analysis. It contains 18,962 training samples and 4,743 test samples. In our experiments, we construct a 5-way ethnicity classification task from the provided ethnicity labels and use the 5 ethnicity categories as subgroups.

- **FER-2013** (Goodfellow et al., 2013) is a facial expression recognition benchmark collected for a representation learning challenge. It contains 35,887 grayscale face images labeled by 7 discrete emotion categories, split into 28,709 images for training and 7,178 for testing. We use the 7 emotion categories as subgroups.

- **GTSRB** (Stallkamp et al., 2011) is a traffic sign recognition dataset containing real-world images of road signs captured under varying illumination, viewpoint, and background conditions. It consists of 51,839 images spanning 43 traffic sign classes, with 39,209 used for training and 12,630 held out for testing. We use the 43 traffic sign classes as subgroups.

- **SVHN** (Netzer et al., 2011) is a digit classification dataset cropped from Google Street View house-number images. It contains color images with substantial background clutter compared to MNIST-style digits and is commonly used as a challenging digit recognition benchmark. There are 10 classes in total, with 73,257 training samples and 26,032 test samples. We use the 10 digit classes as subgroups.

- **CINIC10** (Darlow et al., 2018) is a 10-class image classification dataset that combines images from CIFAR-10 and a subset of ImageNet, yielding higher intra-class diversity and more varied visual conditions than CIFAR-10. It is provided with 90,000 images for training and an additional 90,000 images for testing. We use the 10 object classes as subgroups.

- **CIFAR10** (Krizhevsky et al., 2009) is a canonical object recognition benchmark with 10 categories of natural images. Under the default torchvision (Paszke et al., 2019) split, CIFAR-10 is partitioned into 50,000 training images and 10,000 test images. We use the 10 object classes as subgroups.

### B.2. Baseline and Implementation Details

We compare FairMerging with the following representative task-vector-based merging baselines.

- **Individual** evaluates each task using its own task-specific fine-tuned checkpoint without any merging. This setting serves as a reference point that avoids cross-task interference, while it does not produce a single unified model that can serve all tasks simultaneously.

- **Task Arithmetic** (Ilharco et al., 2023) constructs the merged model by directly adding a weighted sum of task vectors to the shared pretrained backbone. In our six-task setup (UTK-Face, FER-2013, GTSRB, SVHN, CINIC10, CIFAR10), we set the task-wise merging coefficients according to a fixed schedule. Specifically, we use $\alpha = 0.6$ for UTK-Face, $\alpha = 0.4$ for FER-2013, $\alpha = 0.4$ for GTSRB, and $\alpha = 0.5$ for each of SVHN, CINIC10, and CIFAR10. All other implementation details follow the standard Task Arithmetic pipeline (Ilharco et al., 2023).

- **Ties-Merging** (Yadav et al., 2023) mitigates parameter interference by (i) trimming low-magnitude updates, (ii) resolving sign conflicts, and (iii) merging only sign-consistent parameters. We follow the original procedure in Yadav et al. (2023) and set the keep ratio to $0.2$, meaning that at the trimming stage we retain the top $20\%$ entries (by magnitude) of each task vector and discard the rest before performing sign election and disjoint merge. We apply sign election and disjoint merge exactly as specified in Yadav et al. (2023).

- **DARE** (Yu et al., 2024) reduces redundancy in the task vectors by randomly dropping a large fraction of task vector entries and rescaling the remaining updates before applying standard merging. We follow the DARE pipeline in Yu et al. (2024) and set the drop ratio to $0.8$, meaning that for each task vector we stochastically zero out $80\%$ of its entries and retain the remaining $20\%$. After dropping, we apply the prescribed rescaling so that the expected update magnitude is preserved, and then merge the resulting sparse task vectors via a weighted summation on top of the shared pretrained backbone. In our six-task setting, we use task-wise coefficients $\alpha = 0.46$ for UTK-Face, $\alpha = 0.36$ for FER-2013, $\alpha = 0.33$ for GTSRB, $\alpha = 0.12$ for SVHN, $\alpha = 0.46$ for CINIC10, and $\alpha = 0.01$ for CIFAR10.

- **MetaGPT** (Zhou et al., 2024) derives a closed-form rule for selecting the task-arithmetic scaling coefficients by minimizing an upper bound of the average loss difference. Given task checkpoints $\{\boldsymbol{\theta}_t\}_{t=1}^T$ that share the same pretrained initialization $\boldsymbol{\theta}_{\mathrm{pre}}$, it assigns each coefficient proportional to the squared distance from the pretrained weights,

$$\lambda_t = \frac{\|\boldsymbol{\theta}_t - \boldsymbol{\theta}_{\mathrm{pre}}\|_2^2}{\sum_{k=1}^T \|\boldsymbol{\theta}_k - \boldsymbol{\theta}_{\mathrm{pre}}\|_2^2} = \frac{\|\boldsymbol{\tau}_t\|_2^2}{\sum_{k=1}^T \|\boldsymbol{\tau}_k\|_2^2}, \tag{32}$$

where $\boldsymbol{\tau}_t = \boldsymbol{\theta}_t - \boldsymbol{\theta}_{\mathrm{pre}}$ is the task vector. The merged model is then obtained as $\boldsymbol{\theta}_M = \boldsymbol{\theta}_{\mathrm{pre}} + \sum_{t=1}^T \lambda_t \boldsymbol{\tau}_t$.

- **AdaMerging** (Yang et al., 2024) learns merging coefficients by minimizing an unlabeled entropy objective across tasks. Concretely, we optimize the coefficients for 100 steps, and at each step we estimate the entropy objective using 64 mini-batches per task from unlabeled data. We set the temperature in the entropy computation to 1.0 and optimize the coefficients using Adam with a learning rate of 0.01, following the standard practice in Yang et al. (2024).

- **FairMerging (Ours)** consists of two stages that are directly motivated by the fairness bound in Theorem 4.2. Stage 1 reduces the global sensitivity terms, while Stage 2 explicitly controls the merging magnitude term through fairness-aware coefficient optimization. **Stage 1**: Starting from the fine-tuned target checkpoint, we perform an additional fine-tuning step on the target task using the objective in Eqn. (13), which augments the standard subgroup-aggregated loss with a penalty on the across-subgroup variance of gradient norms (to reduce global sensitivity) and an $\ell_2$ proximity term to prevent drifting too far from the original target checkpoint. In practice, we set the weight on the gradient-variance regularizer to $\mu_G = 0.01$, set the curvature regularizer weight to $\mu_\Lambda = 0.01$, and set the proximity regularization coefficient to $\rho = 0.001$. We optimize this Stage 1 objective using the same training pipeline as standard fine-tuning, and use a learning rate of $1 \times 10^{-5}$ for 5 epochs. **Stage 2**: We first orthogonally normalize the task vectors using Gram–Schmidt as in Eq. (14)–(15), and then optimize the merging coefficients by minimizing the sum of the unlabeled entropy loss (Eq. (17)) and the fairness cost (Eq. (19)). We run $50 - 100$ optimization steps, and at each step we evaluate the entropy objective using 64 mini-batches per task, with temperature fixed to 1.0. We optimize the coefficients with Adam using a learning rate of 0.02, and tune the fairness regularization weight $\lambda$ over the range $[1 \times 10^{-5}, 1 \times 10^{-4}]$ across all experiments.

## C. Additional Experimental Results

This section presents complementary experimental evidence to strengthen the main claims. We first provide additional results supporting the empirical observation in Figure 1. We then report extra analyses validating the determinant-based characterization in Figure 2. Finally, we include extended evaluations and ablations for FairMerging beyond what is reported in the main paper.

### C.1. Additional Results Supporting the Observation in Figure 1

In the main paper, Figure 1 demonstrates that standard task-vector-based merging baselines can amplify subgroup performance disparities, even when the individual fine-tuned checkpoints exhibit moderate subgroup gaps. To verify that this phenomenon is not specific to a particular architecture, we replicate the same six-task merging protocol on additional pretrained backbones, including ViT-L/14 and ViT-S/16. For each backbone, we fine-tune one checkpoint per task and then apply Task Arithmetic, Ties-Merging, DARE, MetaGPT, and AdaMerging to produce merged models. We report the maximum accuracy gap across subgroups for each task, using the same subgroup definition as in the main experiments. The detailed results are shown in Figure 3 and Figure 4. Across both additional backbones, we observe the same qualitative pattern as in Figure 1. Model merging consistently increases subgroup performance gaps compared with `Individual`, and the extent of amplification varies markedly across different merging rules.

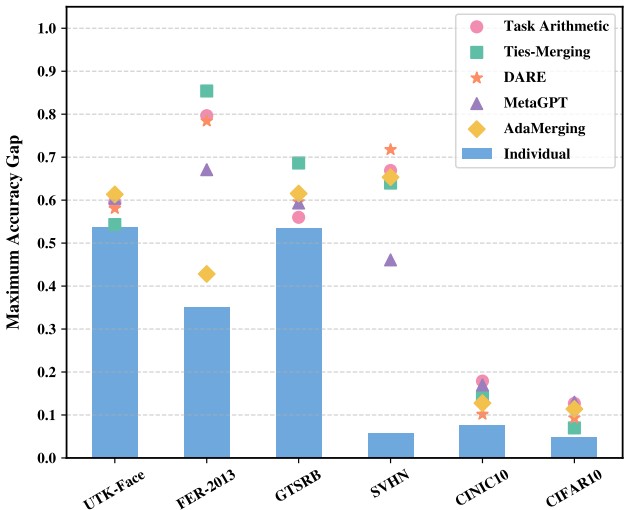

*Figure 3.* Maximum accuracy gap across subgroups before and after model merging on ViT-L/14.

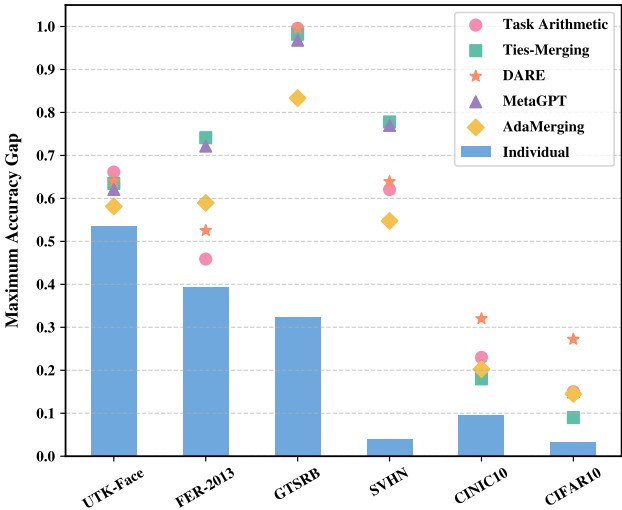

*Figure 4.* Maximum accuracy gap across subgroups before and after model merging on ViT-S/16.

## C.2. Additional Results Supporting the Analysis in Figure 2

Figure 2 in the main text reports a determinant analysis of the merging magnitude $\Delta_\alpha$ and studies how each determinant affects subgroup fairness under model merging on a ViT-B/32 backbone. To verify that the observed trends are not specific to this architecture, we repeat the same controlled sweeps on two additional backbones, ViT-L/14 and ViT-S/16. The full results are reported in Figure 5 and Figure 6.

Across both backbones, the results are consistent with the conclusions obtained on ViT-B/32. First, increasing the number of auxiliary tasks $T$ consistently increases $\Delta_\alpha$ and correspondingly enlarges the measured subgroup disparity, indicating that adding more auxiliary checkpoints tends to introduce a stronger parameter deviation and amplifies fairness risks. Second, decreasing the target coefficient $\alpha_0$ systematically worsens fairness, whereas maintaining a larger $\alpha_0$ yields smaller subgroup performance gaps, supporting the role of the target anchor in controlling merging-induced deviation. Third, we observe a monotonic relationship between the auxiliary coefficients $\{\alpha_t\}_{t=1}^T$ and the fairness metric, with more aggressive auxiliary weighting leading to larger $\Delta_\alpha$ and larger $\mathcal{F}(\boldsymbol{D})$. Finally, enlarging auxiliary task vector norms $\{\|\boldsymbol{\tau}_t\|\}_{t=1}^T$ produces the same effect, confirming that auxiliary updates with larger magnitude contribute disproportionately to fairness degradation when merged without careful control.

These additional experiments demonstrate that the determinant–fairness relationship predicted by Theorem 4.2 is robust across model backbones. In particular, the empirical evidence on ViT-L/14 and ViT-S/16 consistently supports the main claim that $\Delta_\alpha$ serves as a reliable proxy for the severity of merging-induced subgroup disparities, and that manipulating its four determinants provides an effective handle for fairness-aware merging design.

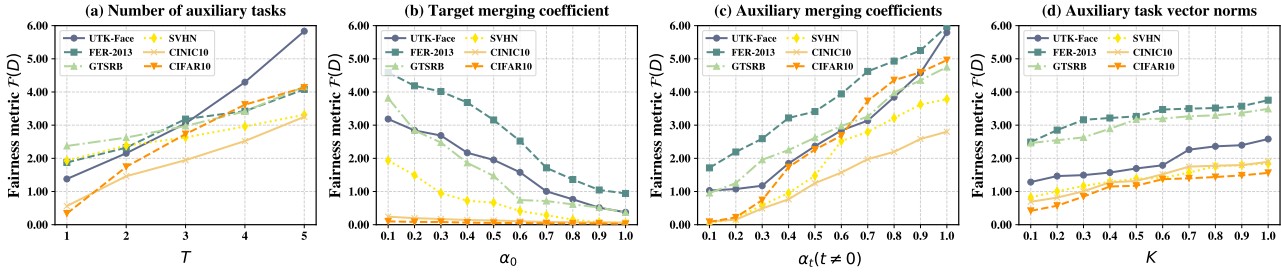

*Figure 5.* Effects of the four determinants of the merging magnitude $\Delta_\alpha$ on the fairness gap $\mathcal{F}(\boldsymbol{D})$. We use ViT-L/14 models fine-tuned on six vision benchmarks, and each curve corresponds to treating one dataset as the target task while merging the remaining ones as auxiliaries. (a) Varying the number of auxiliary tasks $T$. (b–d) With all six tasks included in merging, we vary: (b) the target merging coefficient $\alpha_0$ only; (c) the auxiliary coefficients jointly, using a shared $\alpha_{t\neq0}$ for all auxiliary task vectors; and (d) the auxiliary task vector norms via top-$K$ magnitude masking applied to each auxiliary task vector, following Ties-Merging (Yadav et al., 2023), where $K$ denotes the kept parameter ratio.

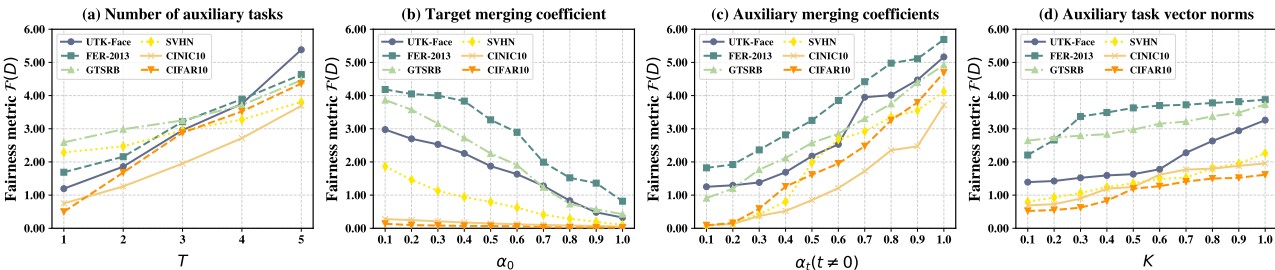

*Figure 6.* Effects of the four determinants of the merging magnitude $\Delta_\alpha$ on the fairness gap $\mathcal{F}(\boldsymbol{D})$. We use ViT-S/16 models fine-tuned on six vision benchmarks, and each curve corresponds to treating one dataset as the target task while merging the remaining ones as auxiliaries. (a) Varying the number of auxiliary tasks $T$. (b–d) With all six tasks included in merging, we vary: (b) the target merging coefficient $\alpha_0$ only; (c) the auxiliary coefficients jointly, using a shared $\alpha_{t\neq 0}$ for all auxiliary task vectors; and (d) the auxiliary task vector norms via top-$K$ magnitude masking applied to each auxiliary task vector, following Ties-Merging (Yadav et al., 2023), where $K$ denotes the kept parameter ratio.

## C.3. Additional Results Supporting the Evaluation of FairMerging

In this section, we provide additional results on two alternative backbones, ViT-L/14 and ViT-S/16, to verify that FairMerging can mitigate fairness degradation while maintaining competitive multi-task performance beyond the default ViT-B/32 setting. We independently fine-tune task-specific models for all six tasks from the same pretrained checkpoint, construct task vectors with respect to the shared pretrained backbone, and compare FairMerging against representative task-vector-based merging baselines (Task Arithmetic, Ties-Merging, DARE, MetaGPT, and AdaMerging).

**FairMerging consistently mitigates unfairness on ViT-L/14 and ViT-S/16.** Tables 3 and 5 report $\mathcal{F}(\boldsymbol{D})$ on each target task under different merging rules. Across both ViT-L/14 and ViT-S/16, we observe the same trend as in Table 1. Specifically, existing merging baselines consistently incur larger subgroup fairness gaps, indicating that task-vector-based merging can amplify subgroup disparities. In contrast, FairMerging achieves the lowest average fairness gap across target tasks, demonstrating that its fairness mitigation effect generalizes across backbones with different capacities.

*Table 3.* Fairness metric $\mathcal{F}(\boldsymbol{D})$ when merging ViT-L/14 models on six tasks.

| Method | UTK-Face | FER-2013 | GTSRB | SVHN | CINIC10 | CIFAR10 | Avg |
|---|---|---|---|---|---|---|---|
| Task Arithmetic (Ilharco et al., 2023) | 1.98 | 2.15 | 3.38 | 2.34 | 0.73 | 0.36 | 1.82 |
| Ties-Merging (Yadav et al., 2023) | 1.67 | 3.18 | 3.71 | 1.49 | 0.27 | 0.26 | 1.76 |
| DARE (Yu et al., 2024) | 1.53 | 2.87 | 2.66 | 1.57 | 0.32 | 0.17 | 1.52 |
| MetaGPT (Zhou et al., 2024) | 1.49 | 1.58 | 4.03 | 1.61 | 0.68 | 0.29 | 1.61 |
| AdaMerging (Yang et al., 2024) | 2.14 | 1.46 | 3.95 | 1.72 | 0.55 | 0.26 | 1.68 |
| FairMerging (Full) | **0.53** | **0.27** | **0.65** | **0.34** | **0.12** | **0.08** | **0.33** |
| Ablation Studies | | | | | | | |
| FairMerging (Stage 1) + Task Arithmetic | 1.54 | 1.92 | 2.84 | 2.15 | 0.64 | 0.30 | 1.57 |
| FairMerging (Stage 1) + Ties-Merging | 1.43 | 2.76 | 3.33 | 1.25 | 0.21 | 0.19 | 1.53 |
| FairMerging (Stage 1) + DARE | 1.26 | 2.39 | 2.12 | 1.38 | 0.27 | 0.12 | 1.26 |
| FairMerging (Stage 1) + MetaGPT | 1.30 | 1.16 | 3.52 | 1.46 | 0.37 | 0.18 | 1.33 |
| FairMerging (Stage 1) + AdaMerging | 1.75 | 1.26 | 3.41 | 1.58 | 0.32 | 0.18 | 1.42 |
| FairMerging (Stage 2) | 0.92 | 0.66 | 0.98 | 0.67 | 0.15 | 0.11 | 0.58 |

**Competitive multi-task performance is preserved across backbones.** Tables 4 and 6 report the corresponding multi-task accuracy. FairMerging remains competitive in Avg Acc while substantially improving subgroup fairness, and it typically matches or outperforms most baselines on a majority of tasks. While AdaMerging can achieve the highest Avg Acc in some settings, FairMerging provides a favorable trade-off by retaining strong multi-task performance under fairness-aware merging.

Overall, the additional results on ViT-L/14 and ViT-S/16 corroborate the main conclusion that FairMerging reliably reduces subgroup unfairness without sacrificing competitive multi-task accuracy, and the observed benefits are not specific to a single backbone.

*Table 4.* Multi-task performance (%) when merging ViT-L/14 models on six tasks.

| Method | UTK-Face | FER-2013 | GTSRB | SVHN | CINIC10 | CIFAR10 | Avg Acc |
|---|---|---|---|---|---|---|---|
| Pretrained | 19.99 | 13.26 | 22.50 | 32.18 | 28.48 | 19.83 | 22.71 |
| Individual | 88.17 | 73.80 | 92.71 | 96.51 | 95.57 | 97.95 | 90.79 |
| Task Arithmetic (Ilharco et al., 2023) | 67.29 | 49.75 | 65.99 | 64.83 | 70.15 | 83.69 | 66.95 |
| Ties-Merging (Yadav et al., 2023) | 70.40 | **69.05** | 71.80 | 69.47 | 84.37 | 90.38 | 75.91 |
| DARE (Yu et al., 2024) | 74.85 | 41.74 | 67.17 | **82.00** | **92.60** | **97.73** | 76.02 |
| MetaGPT (Zhou et al., 2024) | 72.45 | 65.09 | 74.43 | 81.35 | 86.97 | 90.66 | 78.49 |
| AdaMerging (Yang et al., 2024) | 75.69 | 63.95 | **89.53** | 78.54 | 82.21 | 92.83 | **80.46** |
| FairMerging (Full) | **77.86** | 65.47 | 75.72 | 73.46 | 86.33 | 89.48 | 78.05 |

*Table 5.* Fairness metric $\mathcal{F}(\boldsymbol{D})$ when merging ViT-S/16 models on six tasks.

| Method | UTK-Face | FER-2013 | GTSRB | SVHN | CINIC10 | CIFAR10 | Avg |
|---|---|---|---|---|---|---|---|
| Task Arithmetic (Ilharco et al., 2023) | 1.42 | 2.51 | 3.46 | 1.62 | 0.60 | 0.39 | 1.67 |
| Ties-Merging (Yadav et al., 2023) | 1.72 | 3.52 | 3.64 | 1.97 | 0.38 | 0.19 | 1.90 |
| DARE (Yu et al., 2024) | 1.24 | 3.39 | 3.55 | 1.47 | 0.85 | 0.76 | 1.88 |
| MetaGPT (Zhou et al., 2024) | 2.04 | 3.85 | 3.93 | 2.14 | 0.62 | 0.37 | 2.16 |
| AdaMerging (Yang et al., 2024) | 1.94 | 2.27 | 2.36 | 1.40 | 0.48 | 0.32 | 1.46 |
| FairMerging (Full) | **0.68** | **0.51** | **0.39** | **0.42** | **0.18** | **0.05** | **0.37** |
| *Ablation Studies* | | | | | | | |
| FairMerging (Stage 1) + Task Arithmetic | 1.20 | 2.14 | 3.19 | 1.48 | 0.51 | 0.17 | 1.45 |
| FairMerging (Stage 1) + Ties-Merging | 1.36 | 2.97 | 3.23 | 1.65 | 0.20 | 0.13 | 1.59 |
| FairMerging (Stage 1) + DARE | 1.03 | 2.88 | 3.31 | 1.25 | 0.57 | 0.52 | 1.59 |
| FairMerging (Stage 1) + MetaGPT | 1.76 | 3.26 | 3.18 | 1.84 | 0.43 | 0.18 | 1.78 |
| FairMerging (Stage 1) + AdaMerging | 1.65 | 1.98 | 2.85 | 1.16 | 0.23 | 0.20 | 1.35 |
| FairMerging (Stage 2) | 0.92 | 0.88 | 0.73 | 0.71 | 0.22 | 0.10 | 0.59 |

## C.4. Additional Ablation Studies for FairMerging

**Module ablations on other ViT backbones.** To validate the robustness of our two-stage design across models with different capacities, we further reproduced the experiments on ViT-L/14 and ViT-S/16, and report the results in Table 3 and Table 5, consistent with the ablations presented in the main paper. Overall, the results on these additional backbones confirm the conclusions drawn from ViT-B/32: *Stage 1*, as a preprocessing step for the target model, consistently alleviates fairness degradation across various baselines, while the fairness-aware coefficient optimization in *Stage 2* provides more substantial improvements. When combined, the *Full* version achieves the most significant fairness gains. For example, on ViT-L/14, the average fairness metric for the standard merging baselines ranges from 1.52 to 1.82, whereas FairMerging (Full) reduces the average $F(D)$ to 0.33. Stage 1, when combined with the five baselines, consistently shows improvements, e.g., Task Arithmetic decreases from 1.82 to 1.57 and AdaMerging from 1.68 to 1.42, demonstrating that reducing the global sensitivity of the target model serves as a general "dampener" to mitigate differential impacts on subgroups. Using Stage 2 alone also lowers the average $F(D)$ to 0.58, but still lags behind the Full version, highlighting the complementary roles of sensitivity control and fairness cost regularization in practice.

**Ablation on sensitivity-control coefficients in FairMerging Stage 1 ($\mu = \mu_G = \mu_\Lambda$).** We investigate how the sensitivity-control strength in Stage 1 shapes the trade-off between fairness mitigation and task performance by tying the gradient-variance and curvature-variance regularizers with a single coefficient, i.e., $\mu = \mu_G = \mu_\Lambda$. Figures 7–9 report per-target-task

*Table 6.* Multi-task performance (%) when merging ViT-S/16 models on six tasks.

| Method | UTK-Face | FER-2013 | GTSRB | SVHN | CINIC10 | CIFAR10 | Avg Acc |
|---|---|---|---|---|---|---|---|
| Pretrained | 31.76 | 12.25 | 42.37 | 59.26 | 58.59 | 58.14 | 42.73 |
| Individual | 84.78 | 70.02 | 98.03 | 96.63 | 94.19 | 98.69 | 90.39 |
| Task Arithmetic (Ilharco et al., 2023) | 73.81 | 37.80 | 69.79 | 75.55 | 87.13 | 93.94 | 73.00 |
| Ties-Merging (Yadav et al., 2023) | 71.64 | 47.46 | 79.90 | 70.43 | 89.40 | **96.58** | 75.90 |
| DARE (Yu et al., 2024) | 69.87 | 49.20 | 68.61 | 81.02 | **91.00** | 89.26 | 74.83 |
| MetaGPT (Zhou et al., 2024) | 70.26 | 62.44 | 72.49 | 67.35 | 89.65 | 96.01 | 76.37 |
| AdaMerging (Yang et al., 2024) | **77.23** | 47.36 | **90.02** | 81.65 | 86.76 | 94.33 | **79.56** |
| FairMerging (Full) | 70.29 | **65.37** | 70.21 | **83.20** | 85.11 | 90.85 | 77.51 |

results (six subplots per figure) on three backbones (ViT-S/16, ViT-B/32, and ViT-L/14), where we examine the evolution of the fairness metric $F(D)$ and the corresponding test accuracy as $\mu$ increases. Across target tasks and architectures, increasing $\mu$ consistently reduces $F(D)$, indicating that strengthening sensitivity homogenization alleviates merging-induced disparity. This observation aligns with our bound-based analysis: reducing cross-group heterogeneity in local sensitivity (captured by the variance surrogates of $G(a)$ and $\Lambda(a)$) tightens the fairness gap upper bound. Meanwhile, excessively large $\mu$ typically incurs an accuracy drop, as optimization increasingly prioritizes sensitivity regularization and constraint satisfaction over pure empirical risk minimization. Notably, the curves reveal a clear "knee" region: moderate $\mu$ achieves substantial fairness improvements with limited performance degradation, whereas further increasing $\mu$ yields diminishing returns in $F(D)$ but accelerates the loss in accuracy. The consistency of this pattern across model scales suggests that Stage 1 provides a robust and architecture-agnostic mechanism for preconditioning the target checkpoint toward lower global sensitivity before coefficient learning.

**Ablation on fairness-risk regularization in FairMerging Stage 2 ($\eta$).** We further ablate the fairness-aware coefficient optimization in Stage 2 by sweeping the regularization weight $\eta$ in Eqn. (20), which balances the entropy-based multi-task objective $L_{\text{ent}}(\alpha)$ (Eqn. (17)) against the fairness-risk term $J_{\text{fair}}(\alpha)$ (Eqn. (19)). Figures 10–12 summarize this effect for ViT-S/16, ViT-B/32, and ViT-L/14 by plotting the average fairness metric $F(D)$ over the six target tasks and the corresponding average accuracy over the same tasks. As $\eta$ increases, the average $F(D)$ decreases monotonically, demonstrating that explicitly penalizing the effective merging magnitude $\tilde{\Delta}_\alpha$ (Eqn. (18))—and its interaction with the target's global sensitivity indices—offers direct control over merging-induced fairness risk. However, stronger fairness regularization also shifts the optimum away from performance-favoring coefficients, leading to a gradual reduction in average accuracy. Similar to Stage 1, we observe a favorable intermediate regime where sizable fairness gains are obtained with modest accuracy loss, while overly large $\eta$ yields marginal additional fairness improvements but noticeably larger performance degradation. Overall, this ablation verifies that Stage 2 is not merely heuristic: $\eta$ provides an explicit and stable knob for navigating the fairness–accuracy Pareto frontier by trading off entropy minimization against a principled proxy derived from the fairness bound.

## C.5. Additional Experiments on Datasets Commonly Employed in Subpopulation Robustness Evaluations

To further strengthen the practical relevance of the paper, we extend our original six-task ViT-B/32 merging setup by adding Waterbirds and CelebA to form an eight-task merging setting. The detailed results are shown in Figure 13, Tables 7 and 8, which are consistent with our original findings and further support the conclusions.

## C.6. Experiments on Auxiliary-Task Fairness

To show how the proposed method affect fairness on auxiliary tasks, we conduct experiments on auxiliary-task fairness. Specifically, under the same six-task merging setup as in the main paper, we treat each of the six datasets as the target task in turn, merge all six task vectors accordingly, and then additionally evaluate subgroup fairness on the remaining five auxiliary tasks. The detailed results are shown in Tables 9- 14. The results show that, compared with other merging baselines, FairMerging maintains generally comparable auxiliary-task fairness across the remaining five auxiliary tasks, with slight improvements in several cases.

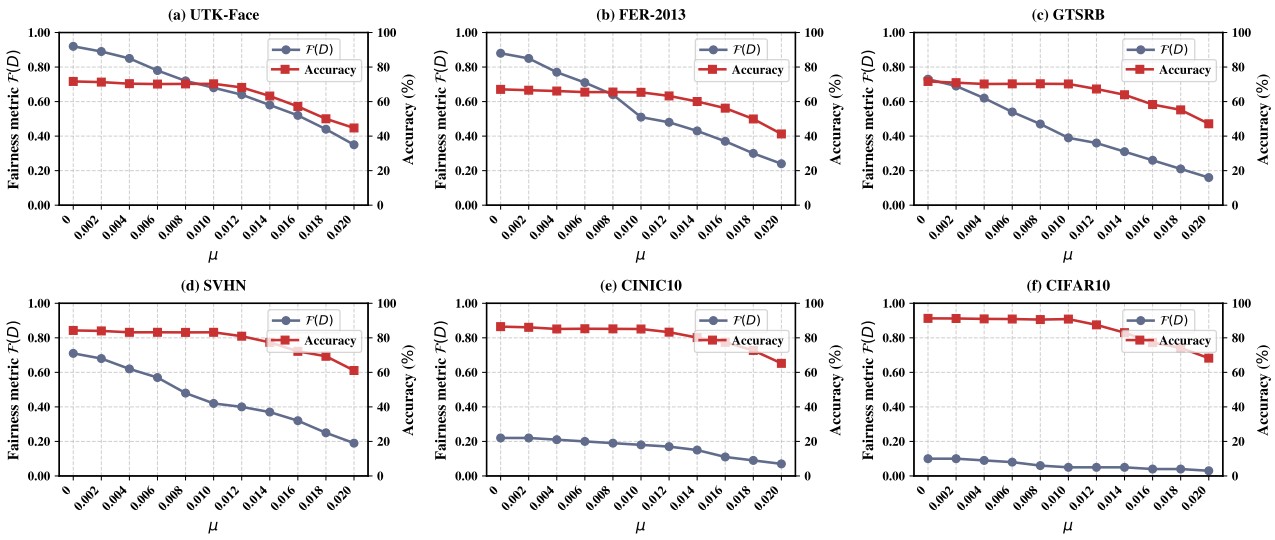

*Figure 7.* Ablation on the sensitivity-control coefficients on ViT-S/16. Following the same six target tasks/datasets (UTK-Face, FER-2013, GTSRB, SVHN, CINIC10, and CIFAR10), we vary the shared coefficient $\mu$ (with $\mu = \mu_G = \mu_\Lambda$) from 0 to 0.02 with a step size of 0.002. For each target task (subfigures (a)–(f)), we report the target-task fairness metric $\mathcal{F}(\boldsymbol{D})$ (left y-axis) together with the target-task test accuracy (%) (right y-axis).

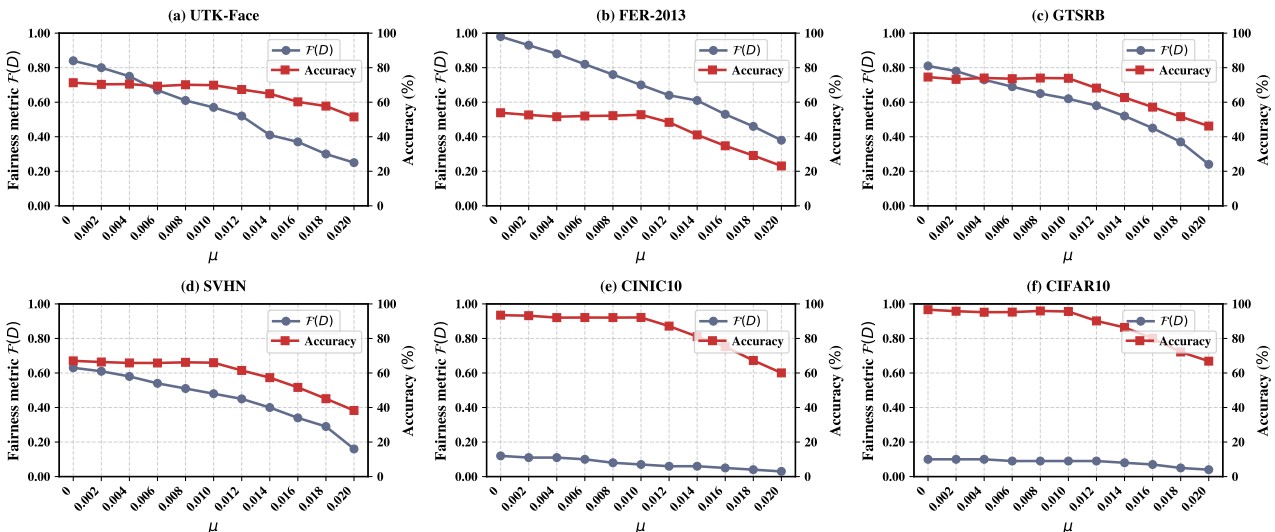

*Figure 8.* Ablation on the sensitivity-control coefficients on ViT-B/32. Following the same six target tasks/datasets (UTK-Face, FER-2013, GTSRB, SVHN, CINIC10, and CIFAR10), we vary the shared coefficient $\mu$ (with $\mu = \mu_G = \mu_\Lambda$) from 0 to 0.02 with a step size of 0.002. For each target task (subfigures (a)–(f)), we report the target-task fairness metric $\mathcal{F}(\boldsymbol{D})$ (left y-axis) together with the target-task test accuracy (%) (right y-axis).

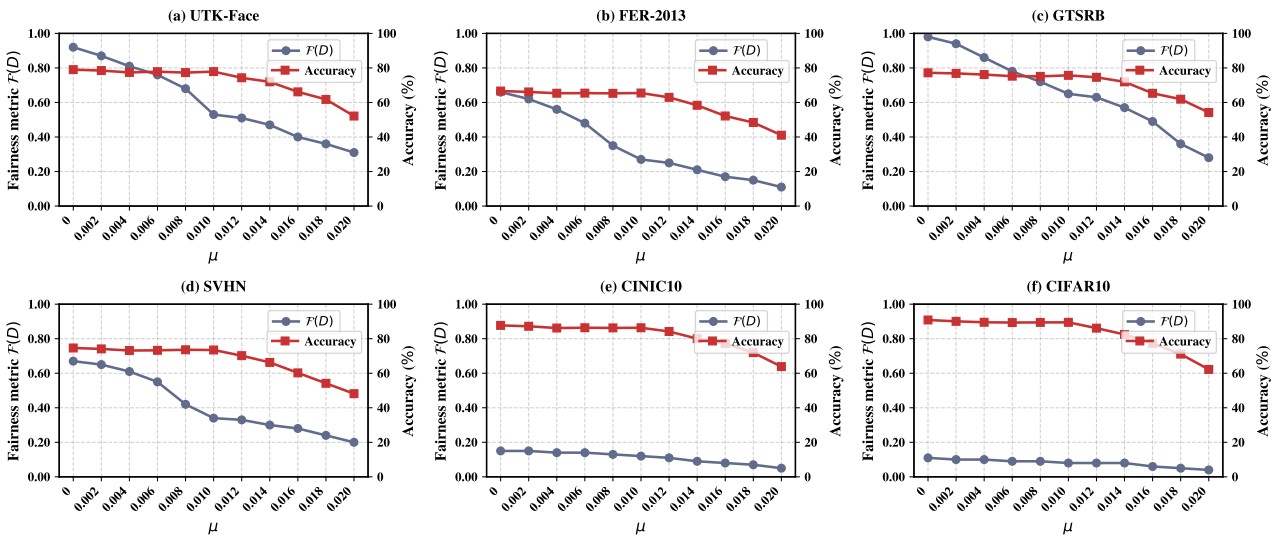

*Figure 9.* Ablation on the sensitivity-control coefficients on ViT-L/14. Following the same six target tasks/datasets (UTK-Face, FER-2013, GTSRB, SVHN, CINIC10, and CIFAR10), we vary the shared coefficient $\mu$ (with $\mu = \mu_G = \mu_\Lambda$) from 0 to 0.02 with a step size of 0.002. For each target task (subfigures (a)–(f)), we report the target-task fairness metric $\mathcal{F}(\boldsymbol{D})$ (left y-axis) together with the target-task test accuracy (%) (right y-axis).

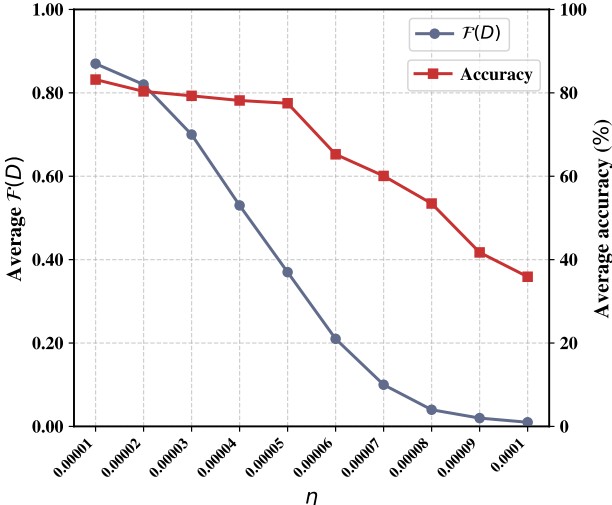

*Figure 10.* Ablation on the fairness-risk regularization weight on ViT-S/16. Following the same six target tasks (UTK-Face, FER-2013, GTSRB, SVHN, CINIC10, and CIFAR10), we sweep the Stage 2 trade-off weight $\eta$ from $1 \times 10^{-5}$ to $1 \times 10^{-4}$ with an increment of $1 \times 10^{-5}$. For each value of $\eta$, we report the average fairness metric $\mathcal{F}(\boldsymbol{D})$ across the six tasks (left y-axis) together with the corresponding average test accuracy (%) across the six tasks (right y-axis).

*Table 7.* Fairness metric $\mathcal{F}(\boldsymbol{D})$ when merging ViT-B/32 models on eight tasks, including the six original tasks and two newly added tasks, CelebA and Waterbirds.

| Method | UTK-Face | FER-2013 | CelebA | Waterbirds | GTSRB | SVHN | CINIC10 | CIFAR10 | Avg |
|---|---|---|---|---|---|---|---|---|---|
| Task Arithmetic | 3.05 | 2.20 | 2.49 | 0.83 | 2.89 | 0.55 | 0.76 | 0.60 | 1.67 |
| Ties-Merging | 2.62 | 2.38 | 1.93 | 1.51 | 3.03 | 2.10 | 0.61 | 0.97 | 1.89 |
| DARE | 2.58 | 2.73 | 3.33 | 0.75 | 4.23 | 0.84 | 1.05 | 0.87 | 2.05 |
| MetaGPT | 3.15 | 2.17 | 2.63 | 1.68 | 2.81 | 0.72 | 0.79 | 0.38 | 1.79 |
| AdaMerging | 2.65 | 2.19 | 1.94 | 1.11 | 2.84 | 2.33 | 0.41 | 0.25 | 1.71 |
| FairMerging (Full) | **0.43** | **0.59** | **0.37** | **0.50** | **0.66** | **0.42** | **0.10** | **0.07** | **0.39** |

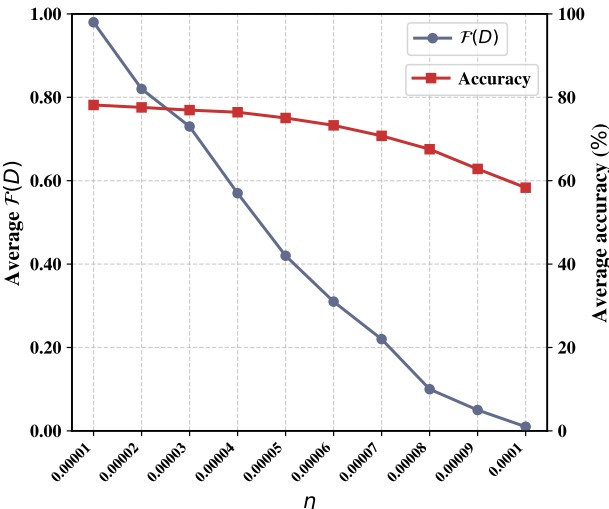

*Figure 11.* Ablation on the fairness-risk regularization weight on ViT-B/32. Following the same six target tasks (UTK-Face, FER-2013, GTSRB, SVHN, CINIC10, and CIFAR10), we sweep the Stage 2 trade-off weight $\eta$ from $1 \times 10^{-5}$ to $1 \times 10^{-4}$ with an increment of $1 \times 10^{-5}$. For each value of $\eta$, we report the average fairness metric $\mathcal{F}(\boldsymbol{D})$ across the six tasks (left y-axis) together with the corresponding average test accuracy (%) across the six tasks (right y-axis).

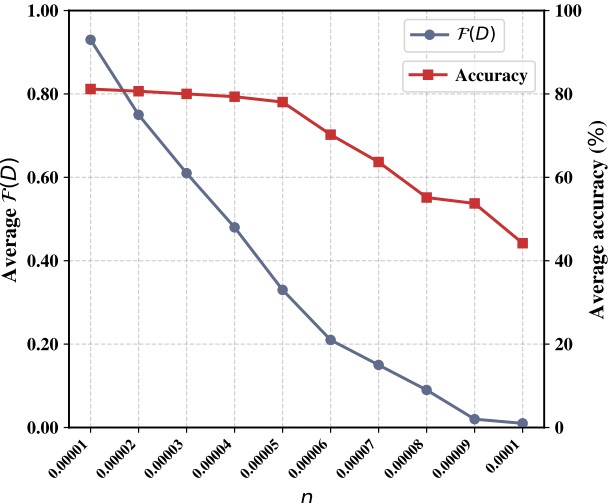

*Figure 12.* Ablation on the fairness-risk regularization weight on ViT-L/14. Following the same six target tasks (UTK-Face, FER-2013, GTSRB, SVHN, CINIC10, and CIFAR10), we sweep the Stage 2 trade-off weight $\eta$ from $1 \times 10^{-5}$ to $1 \times 10^{-4}$ with an increment of $1 \times 10^{-5}$. For each value of $\eta$, we report the average fairness metric $\mathcal{F}(\boldsymbol{D})$ across the six tasks (left y-axis) together with the corresponding average test accuracy (%) across the six tasks (right y-axis).

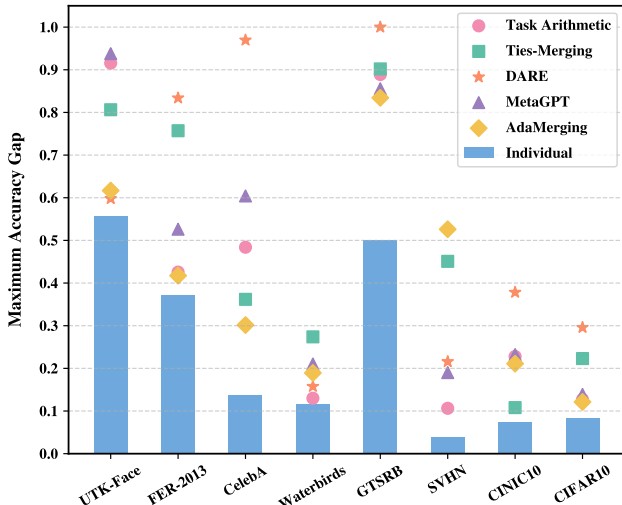

*Figure 13.* Maximum accuracy gap across subgroups before and after model merging on ViT-B/32 across eight datasets, including the six original benchmarks (UTK-Face, FER-2013, GTSRB, SVHN, CINIC10, and CIFAR10) and two newly added datasets, CelebA and Waterbirds.

*Table 8.* Multi-task performance (%) when merging ViT-B/32 models on eight tasks, including the six original tasks and two newly added tasks, CelebA and Waterbirds.

| Method | UTK-Face | FER-2013 | CelebA | Waterbirds | GTSRB | SVHN | CINIC10 | CIFAR10 | Avg Acc |
|---|---|---|---|---|---|---|---|---|---|
| Pretrained | 42.20 | 16.69 | 48.78 | 72.39 | 46.97 | 59.73 | 53.89 | 67.30 | 50.99 |
| Individual | 83.96 | 67.74 | 96.06 | 89.82 | 98.67 | 97.05 | 95.06 | 97.03 | 90.67 |
| Task Arithmetic | 49.25 | 17.84 | 76.35 | 71.19 | 68.13 | 85.01 | 70.27 | 85.33 | 65.42 |
| Ties-Merging | 72.46 | 50.01 | 80.17 | **76.66** | 68.13 | 48.79 | 89.33 | **96.52** | 72.75 |
| DARE | 68.40 | 40.82 | 77.92 | 73.58 | 84.82 | 56.11 | 83.59 | 93.64 | 72.36 |
| MetaGPT | 59.19 | 24.26 | 75.02 | 69.59 | 87.88 | **94.05** | 81.94 | 92.61 | 73.08 |
| AdaMerging | **76.42** | 48.54 | 78.20 | 71.99 | **90.02** | 69.14 | 89.37 | 95.16 | **77.36** |
| FairMerging (Full) | 68.72 | **53.88** | **82.47** | 75.00 | 71.92 | 64.61 | **91.57** | 95.08 | 75.41 |

*Table 9.* Fairness metric $\mathcal{F}(\boldsymbol{D})$ on the auxiliary tasks when merging ViT-B/32 models across six tasks, with UTK-Face treated as the target task.

| Method | FER-2013 | GTSRB | SVHN | CINIC10 | CIFAR10 | Avg |
|---|---|---|---|---|---|---|
| Task Arithmetic | 3.05 | 4.02 | 0.71 | 1.10 | 0.70 | 1.92 |
| Ties-Merging | 4.02 | 3.97 | 1.69 | **0.24** | 0.21 | 2.03 |
| DARE | 2.89 | 3.02 | 3.83 | 0.37 | 0.47 | 2.12 |
| MetaGPT | 2.87 | 2.11 | **0.81** | 0.68 | 0.32 | 1.36 |
| AdaMerging | **2.49** | 2.04 | 1.31 | 0.31 | 0.18 | 1.27 |
| FairMerging (Full) | 2.58 | **1.92** | 1.14 | 0.41 | **0.15** | **1.24** |

*Table 10.* Fairness metric $\mathcal{F}(\boldsymbol{D})$ on the auxiliary tasks when merging ViT-B/32 models across six tasks, with FER-2013 treated as the target task.

| Method | UTK-Face | GTSRB | SVHN | CINIC10 | CIFAR10 | Avg |
|---|---|---|---|---|---|---|
| Task Arithmetic | 3.28 | 4.02 | **0.71** | 1.10 | 0.70 | 1.96 |
| Ties-Merging | 2.52 | 3.97 | 1.69 | **0.24** | 0.21 | 1.73 |
| DARE | 3.71 | 3.02 | 3.83 | 0.37 | 0.47 | 2.28 |
| MetaGPT | 3.15 | 2.11 | 0.81 | 0.68 | 0.32 | 1.41 |
| AdaMerging | 2.62 | **2.04** | 1.31 | 0.31 | 0.18 | **1.29** |
| FairMerging (Full) | **2.37** | 2.46 | 1.28 | 0.55 | **0.17** | 1.37 |

*Table 11.* Fairness metric $\mathcal{F}(\boldsymbol{D})$ on the auxiliary tasks when merging ViT-B/32 models across six tasks, with GTSRB treated as the target task.

| Method | UTK-Face | FER-2013 | SVHN | CINIC10 | CIFAR10 | Avg |
|---|---|---|---|---|---|---|
| Task Arithmetic | 3.28 | 3.05 | **0.71** | 1.10 | 0.70 | 1.77 |
| Ties-Merging | **2.52** | 4.02 | 1.69 | **0.24** | 0.21 | 1.74 |
| DARE | 3.71 | 2.89 | 3.83 | 0.37 | 0.47 | 2.25 |
| MetaGPT | 3.15 | 2.87 | 0.81 | 0.68 | 0.32 | 1.57 |
| AdaMerging | 2.62 | **2.49** | 1.31 | 0.31 | **0.18** | **1.38** |
| FairMerging (Full) | 2.58 | 2.73 | 1.50 | 0.45 | 0.22 | 1.50 |

*Table 12.* Fairness metric $\mathcal{F}(\boldsymbol{D})$ on the auxiliary tasks when merging ViT-B/32 models across six tasks, with SVHN treated as the target task.

| Method | UTK-Face | FER-2013 | GTSRB | CINIC10 | CIFAR10 | Avg |
|---|---|---|---|---|---|---|
| Task Arithmetic | 3.28 | 3.05 | 4.02 | 1.10 | 0.70 | 2.43 |
| Ties-Merging | 2.52 | 4.02 | 3.97 | **0.24** | 0.21 | 2.19 |
| DARE | 3.71 | 2.89 | 3.02 | 0.37 | 0.47 | 2.09 |
| MetaGPT | 3.15 | 2.87 | 2.11 | 0.68 | 0.32 | 1.83 |
| AdaMerging | 2.62 | 2.49 | 2.04 | 0.31 | **0.18** | 1.53 |
| FairMerging (Full) | **2.26** | **2.11** | **2.02** | 0.59 | 0.43 | **1.48** |

*Table 13.* Fairness metric $\mathcal{F}(\boldsymbol{D})$ on the auxiliary tasks when merging ViT-B/32 models across six tasks, with CINIC10 treated as the target task.

| Method | UTK-Face | FER-2013 | GTSRB | SVHN | CIFAR10 | Avg |
|---|---|---|---|---|---|---|
| Task Arithmetic | 3.28 | 3.05 | 4.02 | **0.71** | 0.70 | 2.35 |
| Ties-Merging | **2.52** | 4.02 | 3.97 | 1.69 | 0.21 | 2.48 |
| DARE | 3.71 | 2.89 | 3.02 | 3.83 | 0.47 | 2.78 |
| MetaGPT | 3.15 | 2.87 | 2.11 | 0.81 | 0.32 | 1.85 |
| AdaMerging | 2.62 | 2.49 | **2.04** | 1.31 | **0.18** | **1.73** |
| FairMerging (Full) | 2.81 | **2.25** | 2.46 | 1.33 | 0.28 | 1.83 |

*Table 14.* Fairness metric $\mathcal{F}(\boldsymbol{D})$ on the auxiliary tasks when merging ViT-B/32 models across six tasks, with CIFAR10 treated as the target task.

| Method | UTK-Face | FER-2013 | GTSRB | SVHN | CINIC10 | Avg |
|---|---|---|---|---|---|---|
| Task Arithmetic | 3.28 | 3.05 | 4.02 | **0.71** | 1.10 | 2.43 |
| Ties-Merging | **2.52** | 4.02 | 3.97 | 1.69 | **0.24** | 2.49 |
| DARE | 3.71 | 2.89 | 3.02 | 3.83 | 0.37 | 2.76 |
| MetaGPT | 3.15 | 2.87 | 2.11 | 0.81 | 0.68 | 1.92 |
| AdaMerging | 2.62 | 2.49 | **2.04** | 1.31 | 0.31 | **1.75** |
| FairMerging (Full) | 2.68 | **2.15** | 2.36 | 1.04 | 0.59 | 1.76 |

## C.7. Experiments on Multiple Target Tasks

Our proposed FairMerging can be naturally extended to support fairness intervention for multiple target tasks. Concretely, Stage 1 can be applied separately to each fairness-critical target task to reduce its corresponding global sensitivity before merging. In addition, Stage 2 can be generalized by replacing the current single-target fairness term with a weighted aggregation of fairness costs over multiple target tasks, while keeping the entropy objective unchanged to preserve overall multi-task performance. To further validate this extension, we conduct multi-target fairness experiments under the same six-task merging setup as in the main paper. Concretely, we consider three two-target settings, namely (UTK-Face, FER-2013), (GTSRB, SVHN), and (CINIC10, CIFAR10), and compare FairMerging against the same merging baselines in terms of their fairness performance on these jointly specified target tasks. The detailed results are shown in Table 15, which confirm the effectiveness of FairMerging in jointly improving fairness for more than one target task.

*Table 15.* Fairness metric $\mathcal{F}(\boldsymbol{D})$ when merging ViT-B/32 models across six tasks under three two-target settings: UTK-Face + FER-2013 (yellow region), GTSRB + SVHN (green region), and CINIC10 + CIFAR10 (red region).

| Method | UTK-Face | FER-2013 | GTSRB | SVHN | CINIC10 | CIFAR10 | Avg |
|---|---|---|---|---|---|---|---|
| Task Arithmetic | 3.28 | 3.05 | 4.02 | 0.71 | 1.10 | 0.70 | 2.14 |
| Ties-Merging | 2.52 | 4.02 | 3.97 | 1.69 | 0.24 | 0.21 | 2.11 |
| DARE | 3.71 | 2.89 | 3.02 | 3.83 | 0.37 | 0.47 | 2.38 |
| MetaGPT | 3.15 | 2.87 | 2.11 | 0.81 | 0.68 | 0.32 | 1.66 |
| AdaMerging | 2.62 | 2.49 | 2.04 | 1.31 | 0.31 | 0.18 | 1.49 |
| FairMerging (Full) | **0.64** | **0.79** | **0.68** | **0.55** | **0.11** | **0.13** | **0.48** |

## C.8. Preliminary Experiments on NLP Tasks

Our framework is defined at the level of subgroup-wise losses and therefore does not rely on any vision-specific structure. To examine the generalizability of our findings, we conduct a preliminary fairness study on language classification tasks using RoBERTa-base on three commonly used benchmarks: AG News, Yahoo Answers Topics, and Emotion. The results are shown in Figure 14 and Table 16, which indicate that the fairness trends observed in language tasks echo those in vision tasks, further supporting the generalizability of our findings.

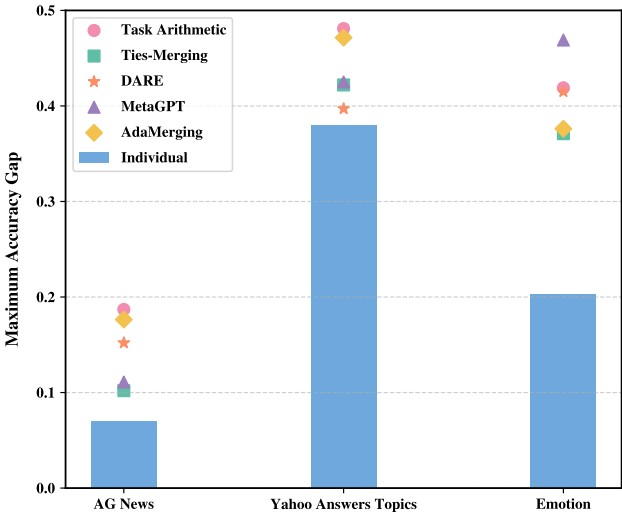

*Figure 14.* Maximum accuracy gap across subgroups before and after model merging on RoBERTa-base for three language classification tasks: AG News, Yahoo Answers Topics, and Emotion.

*Table 16.* Fairness metric $\mathcal{F}(\boldsymbol{D})$ when merging RoBERTa-base models on three language classification tasks: AG News, Yahoo Answers Topics, and Emotion.

| Method | AG News | Yahoo Answers Topics | Emotion | Avg |
|---|---|---|---|---|
| Task Arithmetic | 1.52 | 2.43 | 2.17 | 2.04 |
| Ties-Merging | 1.09 | 3.06 | 1.95 | 2.03 |
| DARE | 1.47 | 2.08 | 2.33 | 1.96 |
| MetaGPT | 1.16 | 2.40 | 2.85 | 2.14 |
| AdaMerging | 1.51 | 2.62 | 1.88 | 2.00 |
| FairMerging (Full) | **0.27** | **0.41** | **0.55** | **0.41** |

