# OpenReview forum: "FairMerging: Rethinking Model Merging through the Lens of Fairness"
_ICML.cc/2026/Conference — ICML 2026 regular_

### Official Review · Reviewer_yWVM · 2026-02-25

**Soundness:** 3
**Presentation:** 4
**Significance:** 3
**Originality:** 3
**Overall Recommendation:** 4
**Confidence:** 3

**Summary:**

In this paper, the authors investigate fairness degradation in model merging and build on their analysis to propose a two-stage fairness-aware merging method. The idea is to refrain fairness as an optimization problem in parameter space during merging, with the goal of reducing subgroup performance gaps while preserving overall performance. The authors validate their approach on multiple vision benchmarks using a ViT-B/32 backbone and further demonstrate consistent mitigation across other transformer variants, strengthening the robustness of architectures and image datasets.

**Compliance With Llm Reviewing Policy:**

Affirmed.

**Final Justification:**

My initial recommendation was positive, and the rebuttal addressed all my concerns, so I maintain my score.

**Key Questions For Authors:**

1. How balanced are the subgroups in each dataset? Could this explain part of the fairness degradation observed?
2. Do your findings generalize to fairness metrics beyond accuracy-based subgroup gaps?
3. Can you provide deeper insight into why certain merging rules amplify subgroup disparities more than others?
4. How does the proposed method scale to scenarios with a large number of tasks?
5. Have you conducted experiments on other modalities beyond vision?

**Limitations:**

The paper would benefit from a deeper discussion of the mechanisms through which merging amplifies subgroup disparities, the implications of relying primarily on accuracy, the scalability of the proposed approach as the number of tasks increases and its generalization beyond vision benchmarks.

**Strengths And Weaknesses:**

Strengths:
- The motivation is well articulated and the research questions are explicitly stated in the manuscript.
- The paper offers a novel perspective by reframing model merging through the lens of subgroup fairness.
- The proposed fairness-aware merging method is structured as a two-stage procedure that is well grounded in the theoretical analysis.
- Experiments span multiple benchmarks and architectures, which reinforces the empirical claims across architectures and image datasets.

Weaknesses:
- The empirical evaluation is limited to vision tasks. Generalization to other domains or modalities is not demonstrated.
- Fairness formulation relies mainly on the accuracy gap, other fairness notions in literature are not explored.
- The scalability of the proposed approach as the number of merged tasks grows is not clear.
- The balance of subgroups is not clearly discussed, leaving open the possibility that dataset imbalance contributes to the observed gaps.

---

> ### Author Rebuttal · Authors · 2026-03-31
>
> Thank you for kindly evaluating that "The paper offers a novel perspective." We also sincerely appreciate your constructive suggestions, and believe that the additional experiments and explanations can address your concerns. The new experimental results are available at https://anonymous.4open.science/r/ppp-6A08/yWVM.pdf.
> - W1&Q5: Thank you for your thoughtful comment on the generalizability of our empirical evaluation beyond vision tasks. We believe that our fairness analysis is equally applicable to other modalities and is not restricted to the vision domain. Our framework is defined at the level of subgroup-wise losses and therefore does not rely on any vision-specific structure. In fact, shortly after submission, we conducted a fairness study on language classification tasks using RoBERTa-base on three commonly used benchmarks: AG News, Yahoo Answers Topics, and Emotion. The results indicate that the fairness trends observed in language tasks echo those in vision tasks, further supporting the generalizability of our findings. To strengthen the empirical scope, we have added the language-task results to the revised manuscript (see Figure 1 and Table 1 in the link).
> - W2&Q2: We sincerely appreciate your thoughtful comment. Generalizing our current analysis to other fairness notions may be nontrivial and could pose additional challenges in identifying the corresponding influencing factors. Nevertheless, as a preliminary exploration, we have examined whether model merging amplifies disparities in Equalized Odds, which requires both the true positive rate and the false positive rate to be comparable across subgroups, and found that it is likewise worsened by model merging (see Figure 2 in the link). We plan to investigate a broader range of fairness metrics in future work to gain a more nuanced understanding of fairness in model merging.
> - W3&Q4: Thank you for your insightful comment regarding the scalability of our method as the number of merged tasks grows. To address this concern, we have added a dedicated task-count ablation in the Appendix. Specifically, we fix UTK-Face as the target task and vary the number of auxiliary tasks from 1 to 5 by progressively including FER-2013, GTSRB, SVHN, CINIC10, and CIFAR10. We then compare the resulting target-task fairness metric under FairMerging and standard merging baselines. The results show that, although the fairness gap increases as more tasks are merged, FairMerging consistently maintains a substantially smaller fairness gap than the corresponding baselines (see Figure 3 in the link). This suggests that the fairness-mitigation effect of FairMerging persists even as the number of merged tasks grows.
> - W4&Q1: Thank you for this important comment. We agree that subgroup imbalance is a plausible factor that may exacerbate the observed fairness gaps. In our submission, we evaluate six datasets across three models, including three imbalanced datasets (UTK-Face, FER-2013, and GTSRB) and three balanced datasets (SVHN, CINIC10, and CIFAR10). As shown in Figures 1, 3, and 4, we observe that the overall fairness gaps tend to be larger on the more imbalanced datasets than on the more balanced ones, suggesting that dataset imbalance may indeed be a contributing factor. We have clarified this point more explicitly in the revised manuscript and consider it an important factor worthy of deeper investigation in future work.
> - Q3: We are grateful for your thoughtful question on why different merging rules affect subgroup disparities differently. A deeper insight is that different merging rules amplify subgroup disparities to different extents because they induce different effective merging magnitudes. According to our theoretical bound in Theorem 4.2, the fairness gap is jointly determined by the merging magnitude and the target model’s global sensitivity. As shown in Figure 2 and discussed in Section 4.3, when the number of merged tasks is fixed, a merging rule that yields a smaller target merging coefficient, larger auxiliary merging coefficients, or larger auxiliary task vector norms will lead to a larger merging magnitude and is therefore more likely to amplify subgroup disparities. We have clarified this point more explicitly in the revised manuscript.
>
> Thank you again for your thorough review, which greatly inspires us to further strengthen the paper. We hope our responses could dispel your concerns and we sincerely hope for your continued support.

---

> > ### Author Rebuttal · Reviewer_yWVM · 2026-03-31
> >
> > The authors have included additional experiments demonstrating the application of their method beyond vision, as well as its scalability according to the number of tasks. The rebuttal answered all my questions.

---

> > > ### Author Response · Authors · 2026-04-08
> > >
> > > We truly appreciate your thoughtful review and constructive suggestions, which have helped us improve the paper. We are also glad to know that our rebuttal has addressed all of your questions. We have incorporated the additional experiments and corresponding discussions into the revised manuscript.

---

### Official Review · Reviewer_mWHm · 2026-03-08

**Soundness:** 3
**Presentation:** 3
**Significance:** 2
**Originality:** 3
**Overall Recommendation:** 4
**Confidence:** 4

**Summary:**

This paper investigates whether task-vector-based model merging amplifies subgroup performance disparities, which has not been systematically studied before. The authors first provide empirical evidence showing that merging consistently widens the maximum accuracy gap across subgroups compared to individual fine-tuned models. They then develop a sensitivity-based theoretical framework and propose FairMerging, a two-stage method that first fine-tunes the target model to reduce global sensitivity via variance penalties on per-subgroup gradient and curvature indices, and then performs fairness-aware coefficient optimization using orthogonally normalized task vectors and an entropy-plus-fairness-cost objective. Experiments show FairMerging reduces average fairness gaps substantially while retaining competitive multi-task accuracy.

**Compliance With Llm Reviewing Policy:**

Affirmed.

**Final Justification:**

My concerns are addressed and I will keep my score

**Key Questions For Authors:**

- Given that model merging is increasingly important for large language models, do you have any results showing that the same situation occurs in language settings? Also, it is critical to well define the fairness in language settings.
- How much labeled data is needed for Stage 1 to be effective? Is the method robust when only a small fraction of labeled target-task data is available, or when subgroup labels are noisy?

**Limitations:**

yes

**Strengths And Weaknesses:**

**Strengths**

- The observation that model merging can systematically amplify subgroup disparities is new and interesting. The paper provides convincing empirical evidence across multiple backbones and merging methods.
- The sensitivity-based decomposition of the fairness bound into merging magnitude and global sensitivity is intuitive. The identification of four determinants of the merging magnitude is useful and each is validated empirically.
- The paper evaluates on six datasets, three backbones, and five baselines, with comprehensive ablations decomposing the contributions of Stage 1 and Stage 2.
- Each component of FairMerging is directly motivated by terms in the theoretical bound, creating a coherent application from theory to algorithm. The use of orthogonal normalization to control auxiliary task vector norms and the variance-based surrogate for global sensitivity are reasonable.
- The paper is generally well-written, with a logical flow from observation, theory, method, to experiments.

**Weaknesses**

- The paper defines subgroups as class labels, which is not a standard notion of fairness. In most fairness literature, subgroups correspond to protected attributes that cut across classes. Equating classes with subgroups conflates task difficulty with fairness, and it is unclear whether the findings generalize to the more standard setting where demographic attributes define subgroups within or across classes.
- The paper acknowledges the absence of language tasks as future work, but given that model merging is heavily used for LLMs, and several cited works are LLM-focused, this limits the significance of the contribution. Additionally, the six datasets are relatively small-scale.
- The pre-merge fine-tuning step needs access to subgroup-level gradients and Hessian information, which requires labeled data and knowledge of subgroup membership. This partially undermines the data-free appeal of model merging and is not discussed as a limitation.
- Using the maximum pairwise difference in excessive losses can be dominated by a single problematic subgroup. For datasets with many classes (e.g., GTSRB with 43 classes), this metric may be noisy. The paper does not discuss the robustness of the metric.
- After Gram-Schmidt orthogonalization and unit-norm rescaling, the task vectors are fundamentally different from the originals. The paper does not analyze whether this transformation degrades the quality of auxiliary task information or how sensitive results are to the ordering of task vectors in the Gram-Schmidt procedure.

---

> ### Author Rebuttal · Authors · 2026-03-31
>
> Great thanks for kindly evaluating that "The observation is new and interesting." We also sincerely appreciate your valuable feedback, and hope that the additional experiments and explanations help address your concerns. The additional experimental results are available at https://anonymous.4open.science/r/ppp-6A08/mWHm.pdf.
> - W1: Thank you for this valuable point. We confirm that our findings can generalize to the setting where demographic attributes define subgroups across classes. We use class-label-defined subgroups for some datasets because class labels provide the most natural and available partition for these benchmarks (e.g., SVHN, CINIC10, and CIFAR10). More generally, our analysis depends on subgroup-wise losses rather than a specific subgroup definition, and therefore applies to arbitrary subgroup partitions, including demographic ones. To further validate this, we have added experiments on datasets with non-class-defined subgroup partitions in the Appendix, including UTK-Face (class label = ethnicity, subgroup = gender), Waterbirds (class label = bird type, subgroup = background), and CelebA (class label = hair color, subgroup = gender). The results align with the central findings of the paper (see Figure 1 and Table 1 in the link).
> - W2&Q1: We sincerely appreciate your insightful comment. Due to space constraints, we refer the reviewer to our response to Reviewer yWVM (W1&Q5), where we provide new language-task experiments and their results. We get similar findings in language settings.
> - W3&Q2: Thanks for your constructive suggestion. To address it, we have added a discussion of this limitation in the revised manuscript. In our implementation, Stage 1 uses 1/5 of the labeled target-task data. To quantify how much labeled data is actually needed, we have added an ablation in the Appendix by varying the labeled-data fraction from 0 to 1. The fairness metric F(D) decreases slightly as the labeled fraction increases, with most improvement already achieved at around 1/6, after which the curve becomes largely stable (see Figure 3 in the link). Moreover, even with only 1/9 of the labeled data, F(D) already achieves about 76% of the fairness improvement obtained at 1/6, suggesting that the method remains effective with a small labeled subset. To further evaluate robustness to noisy subgroup labels, we have added an ablation in the Appendix by replacing each subgroup label with a randomly chosen incorrect one with probability $p$, and $p$ is varied from 0% to 100%. As $p$ increases, the fairness gain weakens. However, when $p$ ranges from 0 to 0.2, FairMerging still maintains a clear fairness advantage over the baselines (see Figure 4 in the link), indicating that the method remains useful under low subgroup-label noise.
> - W4: We sincerely appreciate your thoughtful comment. To assess this issue, we have added a robustness check on GTSRB with ViT-B/32 in the Appendix. For each merging method, we identify the subgroup with the largest excessive loss $E(a)$, remove that subgroup from evaluation, and then recompute the fairness metric on the remaining 42 subgroups. As expected, removing the worst subgroup reduces the fairness gap for all methods. Importantly, FairMerging still achieves the lowest fairness gap among all methods after removal, indicating that our main conclusion is not an artifact of a single problematic subgroup (see Table 3 in the link).
> - W5: Thank you for pointing out the potential impact of the Gram–Schmidt procedure. To address this concern, we have added two additional ablations in the Appendix. First, to assess whether this transformation degrades the quality of auxiliary task, we compare FairMerging with and without this transformation under the same Stage 2 optimization objective. The results show that auxiliary-task performance remains comparable (see Tables 4 and 5 in the link). Second, to evaluate the sensitivity to the ordering of task vectors, we keep the target vector fixed, randomly permute the processing order of the auxiliary task vectors, repeat the procedure 10 times, and report the resulting fairness metrics. The results show that the fairness outcomes are stable across different orderings (see Figure 5 in the link). These observations are also intuitively consistent with our design. The Gram–Schmidt step reduces redundancy among auxiliary task vectors and mitigates the accumulation of correlated updates, while the Stage 2 entropy objective helps preserve overall multi-task performance. As for the ordering issue, different orderings may lead to slightly different vectors, but they all serve the same purpose of reducing cross-task redundancy. After coefficient optimization, these ordering-induced differences remain minor in practice. Taken together, these results suggest that the transformation is both effective and robust.
>
> We highly value your thoughtful feedback and are deeply grateful for your continued support.

---

> > ### Author Rebuttal · Reviewer_mWHm · 2026-04-02
> >
> > Thanks for the response. My concerns are addressed. I will keep my score as it is positive.

---

> > > ### Author Response · Authors · 2026-04-08
> > >
> > > We greatly appreciate your thoughtful review and helpful comments, which have been valuable in improving the paper. It is encouraging to know that our rebuttal has resolved your questions and concerns. We have revised the manuscript accordingly to incorporate the clarifications and additional discussions raised in the rebuttal.

---

### Official Review · Reviewer_PwLH · 2026-03-16

**Soundness:** 4
**Presentation:** 3
**Significance:** 4
**Originality:** 3
**Overall Recommendation:** 5
**Confidence:** 4

**Summary:**

The paper studies subgroup performance disparities arising in model merging. It formally defines the subgroup-level impact of model merging as the excess risk of the merged model relative to the task-specific model. It further defines the fairness impact across subgroups as the maximum pairwise difference in excess risks. The authors derive theoretical upper bounds for both excess risk and fairness disparity. This analysis helps characterize the factors that influence these quantities, such as merging magnitude, subgroup sensitivity, and global sensitivity. Building on these insights, the paper proposes FairMerging, a framework designed to mitigate the impact of these factors. The method is evaluated on six tasks using three backbone models and compared against five merging baselines, with results showing improved fairness while maintaining competitive multi-task performance.

**Compliance With Llm Reviewing Policy:**

Affirmed.

**Final Justification:**

The responses have fully addressed my concerns, and I have updated my overall score as well as my scores on soundness and significance to reflect this.

**Key Questions For Authors:**

1. How does the proposed method affect fairness on auxiliary tasks?
2. Can the proposed method support fairness intervention for more than one target task?

**Strengths And Weaknesses:**

### Soundness
- (+) The claims are supported by extensive empirical experiments as well as theoretical analysis
- (+) An ablation study effectively illustrates the importance of the proposed two-stage approach
- (-) The choice of benchmark datasets makes it somewhat difficult to fully appreciate the practical relevance of the fairness aspects. Using datasets commonly employed in subpopulation robustness evaluations (e.g., Waterbirds, CelebA) could provide more insightful evidence


### Presentation
- (+) The presentation is clearly written and well structured
- (+) The literature review is fairly comprehensive (add more discussions on subgroup robustness literature)
- (-) It is not clearly specified which subgroups are used to evaluate fairness in some of the benchmark datasets considered

### Significance
- (+) The paper contributes to a better understanding of the sources of unfairness in existing model merging interventions and proposes a method to mitigate them
- (-) The paper lacks a discussion of the overhead costs associated with the proposed intervention, particularly in comparison to data-free alternatives. This includes aspects such as additional data requirements, computational cost, and potential impacts on the fairness of auxiliary tasks

### Originality
- (+) The work highlights factors influencing unfairness in existing model merging methods

---

> ### Author Rebuttal · Authors · 2026-03-31
>
> Thank you sincerely for your positive evaluation. We also truly appreciate your constructive suggestions. We have conducted additional experiments and provided further explanations to address your concerns. The additional experimental results are available at https://anonymous.4open.science/r/ppp-6A08/PwLH.pdf.
> - W1: We appreciate your suggestion that benchmarks such as Waterbirds or CelebA would further strengthen the practical relevance of the paper. To address this point, in the revised manuscript, we have extended our original six-task ViT-B/32 merging setup by adding Waterbirds and CelebA to form an eight-task merging setting. We then take Waterbirds and CelebA as the target task, respectively, to examine the resulting fairness changes under model merging. The results are consistent with our original findings and further support the conclusions (see Figure 1, Tables 1 and 2 in the link).
> - W2: Thank you for pointing this out. Concretely, the subgroups are defined as follows: UTK-Face uses the 5 ethnicity categories as subgroups, FER-2013 uses the 7 emotion categories, GTSRB uses the 43 traffic-sign classes, SVHN uses the 10 digit classes, and CINIC10 and CIFAR10 use the 10 object classes. We have revised the manuscript to explicitly list the subgroup definition for each dataset.
> - W3: Thanks for your constructive suggestion. To address it, we have added a dedicated discussion of the overhead costs of our method in the revised manuscript. Concretely, FairMerging introduces two main sources of overhead: (i) Stage 1 performs an additional fine-tuning step on the target task, which requires labeled target-task data and subgroup annotations; and (ii) Stage 2 optimizes the merging coefficients using unlabeled data from all tasks, which introduces additional optimization cost. However, this overhead is moderate in practice: the main additional cost is confined to Stage 1, which only requires 1--5 epochs of fine-tuning, whereas Stage 2 is comparable in spirit and scale to AdaMerging, which also relies on unlabeled entropy minimization.
> - Q1: Thank you for highlighting the important issue of auxiliary-task fairness. We would like to clarify that the current formulation of FairMerging is primarily target-centric. Nevertheless, the Gram–Schmidt orthogonalization step applied to auxiliary task vectors can reduce cross-task redundancy and mitigate interference among auxiliary updates. To further address this concern, we have conducted additional experiments on auxiliary-task fairness in the Appendix. Specifically, under the same six-task merging setup as in the main paper, we treat each of the six datasets as the target task in turn, merge all six task vectors accordingly, and then additionally evaluate subgroup fairness on the remaining five auxiliary tasks. The results show that, compared with other merging baselines, FairMerging maintains generally comparable auxiliary-task fairness across the remaining five auxiliary tasks, with slight improvements in several cases (see Tables 3-8 in the link).
> - Q2: We appreciate your thoughtful comment on extending FairMerging to multiple target tasks. We believe that our proposed FairMerging can be naturally extended to support fairness intervention for multiple target tasks. Concretely, Stage 1 can be applied separately to each fairness-critical target task to reduce its corresponding global sensitivity before merging. In addition, Stage 2 can be generalized by replacing the current single-target fairness term with a weighted aggregation of fairness costs over multiple target tasks, while keeping the entropy objective unchanged to preserve overall multi-task performance. To further validate this extension, we have added multi-target fairness experiments in the Appendix under the same six-task merging setup as in the main paper. Concretely, we consider three two-target settings, namely (UTK-Face, FER-2013), (GTSRB, SVHN), and (CINIC10, CIFAR10), and compare FairMerging against the same merging baselines in terms of their fairness performance on these jointly specified target tasks. The results confirm the effectiveness of FairMerging in jointly improving fairness for more than one target task (see Table 9 in the link).
>
> Your insights are valuable to us. We hope our responses could dispel your concerns and sincerely appreciate your further support.

---

> > ### Author Rebuttal · Reviewer_PwLH · 2026-04-04
> >
> > Thank you for the rebuttal response. The response has fully addressed my concerns, and I will update my overall score as well as my scores on soundness and significance to reflect this.

---

> > > ### Author Response · Authors · 2026-04-08
> > >
> > > We sincerely thank you for your thoughtful review and constructive feedback, which have helped us further strengthen the paper. We are also grateful that our rebuttal has fully addressed your concerns. We have incorporated the clarifications and additional discussions from the rebuttal into the revised manuscript.

---

### Decision · Program_Chairs · 2026-04-30

**Decision:**

Accept (regular)

**Comment:**

This paper studies a subgroup fairness problem, defined as the performance disparities arising in model merging. It derives upper bounds for both excess risk and fairness disparity, which shed light on factors contributing to the disparity, including merging magnitude, subgroup sensitivity, and global sensitivity. A merging framework is then designed to mitigate these impacts.

The three reviews are largely positive, acknowledging the paper's contribution to a better understanding of the sources of unfairness in existing model merging methods. They all found the paper's perspectives novel and interesting. The rebuttal process was helpful, leading to one reviewer incrementing their score.